# TARM1 contributes to development of arthritis by activating dendritic cells through recognition of collagens

Rikio Yabe[1,2], Soo-Hyun Chung [1], Masanori A. Murayama[1], Sachiko Kubo[1], Kenji Shimizu [1], Yukiko Akahori[2], Takumi Maruhashi [1], Akimasa Seno [1], Tomonori Kaifu[1], Shinobu Saijo[2 ✉] & Yoichiro Iwakura [1,2 ✉]

TARM1 is a member of the leukocyte immunoglobulin-like receptor family and stimulates macrophages and neutrophils in vitro by associating with FcRγ. However, the function of this molecule in the regulation of the immune system is unclear. Here, we show that *Tarm1* expression is elevated in the joints of rheumatoid arthritis mouse models, and the development of collagen-induced arthritis (CIA) is suppressed in *Tarm1⁻/⁻* mice. T cell priming against type 2 collagen is suppressed in *Tarm1⁻/⁻* mice and antigen-presenting ability of GM-CSF-induced dendritic cells (GM-DCs) from *Tarm1⁻/⁻* mouse bone marrow cells is impaired. We show that type 2 collagen is a functional ligand for TARM1 on GM-DCs and promotes DC maturation. Furthermore, soluble TARM1-Fc and TARM1-Flag inhibit DC maturation and administration of TARM1-Fc blocks the progression of CIA in mice. These results indicate that TARM1 is an important stimulating factor of dendritic cell maturation and could be a good target for the treatment of autoimmune diseases.

[1] Center for Animal Disease Models, Research Institute for Biomedical Sciences (RIBS), Tokyo University of Science, Noda, Chiba 278-0022, Japan. [2] Medical Mycobiology Research Center, Chiba University, Chiba, Chiba 260-8673, Japan. ✉email: saijo@faculty.chiba-u.jp; iwakura@rs.tus.ac.jp

The leukocyte immunoglobulin (Ig)-like receptor (LILR) family is a member of Ig superfamily[1–3]. Some of LILR family members are expressed in myeloid cells, such as dendritic cells (DCs) and macrophages, and transduce activation or inhibitory signals into these cells[1–3]. Because these family members are closely associated with autoimmune diseases, the importance of these molecules in the homeostasis of the immune system is suggested[4–6].

T cell-interacting, activating receptor on myeloid cells-1 (TARM1; gene symbol *Tarm1*) is a recently identified LILR family member encoded within the leukocyte receptor complex[7]. TARM1 consists of two extracellular Ig-like domains, a trans-membrane domain and a short cytoplasmic tail with no typical signaling motif. However, TARM1 specifically associates with immunoreceptor tyrosine-based activation motif (ITAM)-bearing adapter protein Fc receptor γ-chain to transduce signals[7]. Amino acid alignment analysis indicates that TARM1 is closely related to osteoclast-associated receptor (OSCAR), which is involved in antigen presentation and activation of human DCs[8–10] and differentiation of osteoclasts in mice by binding to specific motifs within fibrillar collagens[11,12]. Radjabova et al. reported that *Tarm1* is expressed constitutively in bone marrow (BM)-derived granulocytes, monocytes, neutrophils, and granulocyte/macrophage colony-stimulating factor (GM-CSF)-induced DCs (GM-DCs) and the expression is upregulated by the stimulation with lipopolysaccharide (LPS) or bacterial challenge[7]. They also reported that TARM1 signaling enhances tumor necrosis factor (TNF) and interleukin (IL)-6 production from macrophages and neutrophils[7]. However, the role of TARM1 in health and disease largely remains to be elucidated.

Rheumatoid arthritis (RA) is a typical autoimmune disease characterized by synovial inflammation and bone destruction[13]. The pathogenic mechanism is complex because multiple factors such as genetic susceptibility and environmental factors are involved in the pathogenesis of RA[13,14]. However, it is widely recognized that autoimmune responses against self-antigens such as joint components and immunoglobulins causes overproduction of inflammatory cytokines such as TNF, IL-6, IL-1, and IL-17 from immune cells and synovial cells, resulting in the synovial inflammation and bone destruction[13–15]. During the development of RA, DCs play crucial roles in the initiation and amplification of immune responses by presenting self-antigens to T cells and producing proinflammatory cytokines[16]. DCs also express various innate immune receptors such as TLRs and C-type lectin receptors that are important for the activation and maturation of DCs[5,17–19].

Many RA models are developed for the study of RA pathogenesis and drug validation[20]. Collagen-induced arthritis (CIA) is one of the most widely used models[21]. In this model, antibodies against type 2 collagen (IIC) play a crucial role for the development of arthritis[22]. However, anti-IIC IgG concentrations in serum do not completely correlate with the severity of arthritis[23], because IIC-specific antibodies contain not only arthritogenic antibodies but also non-arthritogenic antibodies[22,24]. We have generated two mouse models: human T cell leukemia virus type I (HTLV-I)-transgenic (Tg) and IL-1 receptor antagonist (IL-1Ra; gene symbol *Il1rn*)-deficient mice that spontaneously develop arthritis resembling RA[25,26]. By analyzing these models using DNA microarray, we identified a group of genes whose expression is specifically augmented in arthritic mice[27] and analyzed the roles of these genes in the development of arthritis[19,28–31]. *Tarm1* is one of such genes whose expression is augmented in arthritic joints of both HTLV-I Tg and *Il1rn*$^{-/-}$ mouse RA models.

In this study, we demonstrate the role of TARM1 in the development of autoimmune arthritis by generating *Tarm1*$^{-/-}$ mice. *Tarm1*$^{-/-}$ mice are refractory against CIA, and recall proliferative response of T cells against IIC is decreased due to impaired antigen presentation by DCs. In vitro development of mature DCs from BM cells in response to GM-CSF is impaired in *Tarm1*$^{-/-}$ mice. We reveal that IIC, which is expressed on GM-DCs, is an endogenous ligand for TARM1 and controls maturation of GM-DCs. Furthermore, we show that administration of soluble TARM1 attenuates the development of CIA in mice, suggesting that TARM1 is a good target for the treatment of autoimmune diseases.

## Results

**Tarm1$^{-/-}$ mice are refractory to the development of CIA.** Previously, we found that the expression of LILR family members increases in arthritic joints of RA models using microarray analysis, suggesting possible involvement of these molecules in the development of arthritis[27]. Accordingly, we examined the expression of *Tarm1* in joints of HTLV-I-Tg and *Il1rn*$^{-/-}$ mice by quantitative PCR (qPCR) analysis and found that *Tarm1* expression is significantly upregulated in arthritic joints compared with control mouse joints (Supplementary Fig. 1a, b).

Then we investigated the role of TARM1 in the development of autoimmune arthritis using *Tarm1*$^{-/-}$ mice, in which the *Tarm1* gene exon 1 was replaced by enhanced green fluorescence protein (EGFP) and the neomycin-resistant gene by homologous-recombination techniques (Supplementary Fig. 1c–f). *Tarm1*$^{-/-}$ mice were born at the expected Mendelian ratio, grew and bred normally, and showed no obvious abnormalities at least at 1 year of age under specific pathogen-free (SPF) conditions (Supplementary Fig. 1g). Immune cell compositions in lymph nodes (LNs), spleen, and BM were normal in *Tarm1*$^{-/-}$ mice (Supplementary Fig. 1h).

We examined susceptibility of *Tarm1*$^{-/-}$ mice to CIA. The incidence of arthritis in *Tarm1*$^{-/-}$ mice was significantly reduced and the severity score was also markedly attenuated throughout the disease stages compared with wild-type (WT) mice (Fig. 1a–c). Histopathological analysis using hematoxylin and eosin (HE) and Safranin O staining revealed that synovial inflammation, pannus formation, and cartilage and bone destruction are remarkably decreased in *Tarm1*$^{-/-}$ mouse joints (Fig. 1d–f). Flow cytometric analysis revealed that DC (CD11c$^+$), mature DC (I-A/I-E$^+$CD11c$^+$), activated CD4$^+$ T cell (CD44$^+$CD4$^+$), and B cell (CD19$^+$) populations were reduced in inguinal LNs of *Tarm1*$^{-/-}$ mice (Fig. 1g). We also found that IIC-specific serum IgG2a, IgG2b, and IgG3 concentrations were reduced in *Tarm1*$^{-/-}$ mice (Fig. 1h). Although IgG concentrations in some non-arthritic mice were similarly high as arthritic mice, this is probably because IIC-specific antibodies contain non-arthritogenic antibodies in addition to arthritogenic antibodies[22]. These results indicate that *Tarm1*$^{-/-}$ mice are refractory to the induction of CIA compared to WT mice.

**TARM1 is expressed by and is required for the activation of DCs.** Then, we investigated the expression of *Tarm1* among LN cells. By using EGFP expression as the indicator, we found that *Tarm1* was highly expressed in inflammatory-type (I-A/I-E$^+$ Ly6C$^+$CD11b$^+$CD11c$^+$) DCs in draining LNs (dLNs) after induction of CIA in *Tarm1*$^{+/-}$ mice (Fig. 2a). The expression was weakly observed in CD11b$^+$ conventional DCs (cDCs) but not in CD24$^+$ cDCs, B220$^+$ plasmacytoid DCs, macrophages, neutrophils, CD4$^+$ T cells, CD8$^+$ T cells, and B cells under physiological conditions (Supplementary Fig. 2a).

The expression of *Tarm1* was clearly observed in in vitro differentiated CD11c$^+$ GM-DCs and the expression was further enhanced in the inflammatory-type (I-A/I-E$^+$CD11c$^+$CD11b$^+$ Ly6C$^+$) subset of GM-DCs (Fig. 2b), whereas it was only weakly observed in CD11b$^+$ Flt3L-induced DCs (CD11b$^+$ FL-DCs) and

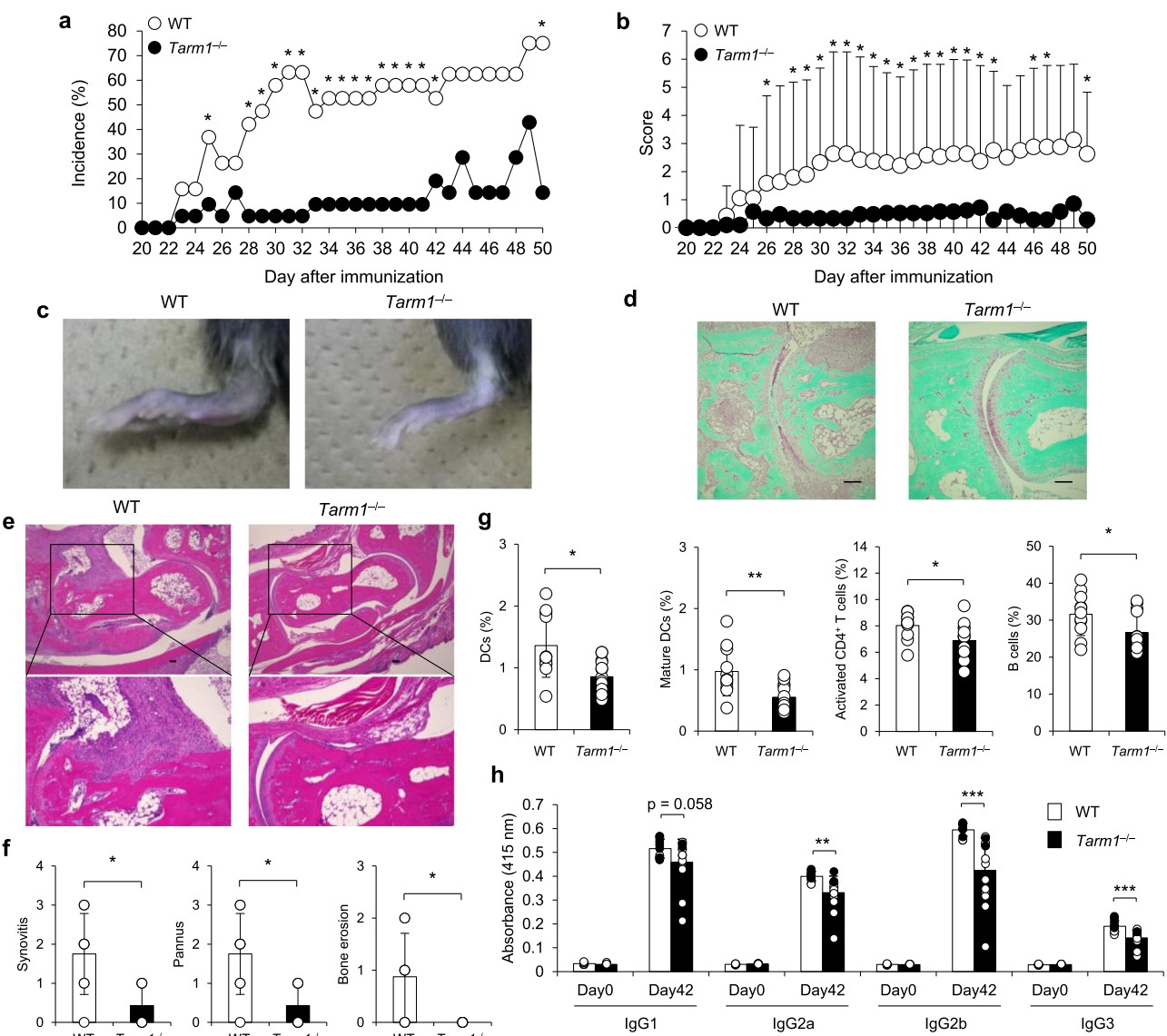

**Fig. 1 Development of CIA is suppressed in *Tarm1*<sup>−/−</sup> mice. a**, **b** Incidence (**a**) and severity score (**b**) of arthritis in WT and *Tarm1*<sup>−/−</sup> mice immunized with IIC plus CFA. The sum of individual score was divided by the total mouse number in **b**. Data from two independent experiments are combined. WT = 19, *Tarm1*<sup>−/−</sup> = 21. Mean ± SD. *P < 0.05 [$\chi^2$-test (**a**) and two-tailed Mann–Whitney U test (**b**)]. **c** Representative images of ankle joints from WT and *Tarm1*<sup>−/−</sup> mice 50 days after CIA induction. **b**–**f** Representative histology of the affected joints after induction of CIA. Sections (5 µm) of hind limbs of WT and *Tarm1*<sup>−/−</sup> mice at day 50 after immunization were stained with Safranin O (**d**) and HE (**e**). Histopathological scores including synovitis, pannus, and bone erosion are shown in (**f**). WT = 8, *Tarm1*<sup>−/−</sup> = 7. Mean ± SD. Scale bar, 100 µm. *P < 0.05 (two-tailed Mann–Whitney U test). **g** Contents of DCs (CD11c<sup>+</sup>), mature DCs (I-A/I-E<sup>+</sup>CD11c<sup>+</sup>), and activated T (CD44<sup>+</sup>CD4<sup>+</sup>) and B (CD19<sup>+</sup>) cells in inguinal LNs from WT and *Tarm1*<sup>−/−</sup> mice at 42 days after IIC immunization were determined by flow cytometry. WT = 11, *Tarm1*<sup>−/−</sup> = 14. Mean ± SD. *P < 0.05; **P < 0.01 (two-tailed unpaired t test). **h** IIC-specific IgGs in sera were determined by ELISA. WT = 11, *Tarm1*<sup>−/−</sup> = 14. Mean ± SD. Closed circle = arthritic mouse, open circle = non-arthritic mouse. **P < 0.01; ***P < 0.001 (two-tailed unpaired t test). Source data are provided as a Source data file.

not in CD24<sup>+</sup> FL-DCs and B220<sup>+</sup> FL-DCs (Supplementary Fig. 2b). EGFP expression was also detected in BM-derived macrophages and BM neutrophils (Supplementary Fig. 2b), although qPCR analysis indicated that *Tarm1* expression in BM macrophages, BM osteoclasts, BM neutrophils, BM monocytes, blood neutrophils, blood monocytes, T cells, and B cells was much lower or not detected compared with GM-DCs (Fig. 2c). *Tarm1* expression in GM-DCs was further elevated after stimulation with inflammatory cytokines (TNF and IL-1β) and pathogen-associated molecular pattern molecules (PAMPs) [LPS, CpG, and poly(I:C)] (Supplementary Fig. 2c).

Because TARM1 is highly expressed in mature inflammatory DCs, we next investigated the roles of TARM1 in the

differentiation of DCs. The efficiency of differentiation of CD11c<sup>+</sup> GM-DCs from BM cells after treatment with GM-CSF was similar between WT and *Tarm1*<sup>−/−</sup> mice (Supplementary Fig. 2d), and the proportion of total CD11c<sup>+</sup> cells was similar between WT and *Tarm1*<sup>−/−</sup> mice (Fig. 2d). However, the proportion of the I-A/I-E<sup>hi</sup>CD11c<sup>+</sup> cell population in *Tarm1*<sup>−/−</sup> mice was lower than that in WT mice (Fig. 2e). The expression of DC maturation markers such as I-A/I-E, CD86, and CD80 were significantly decreased in GM-DCs from *Tarm1*<sup>−/−</sup> mice compared with that from WT mice (Fig. 2f and Supplementary Fig. 2e). DNA microarray analysis revealed that the expression of costimulatory receptor genes such as *Cd70*, *Pvrl2*, *Tnfsf18*, and *Tnfsf4* was decreased in *Tarm1*<sup>−/−</sup> GM-DCs compared with WT

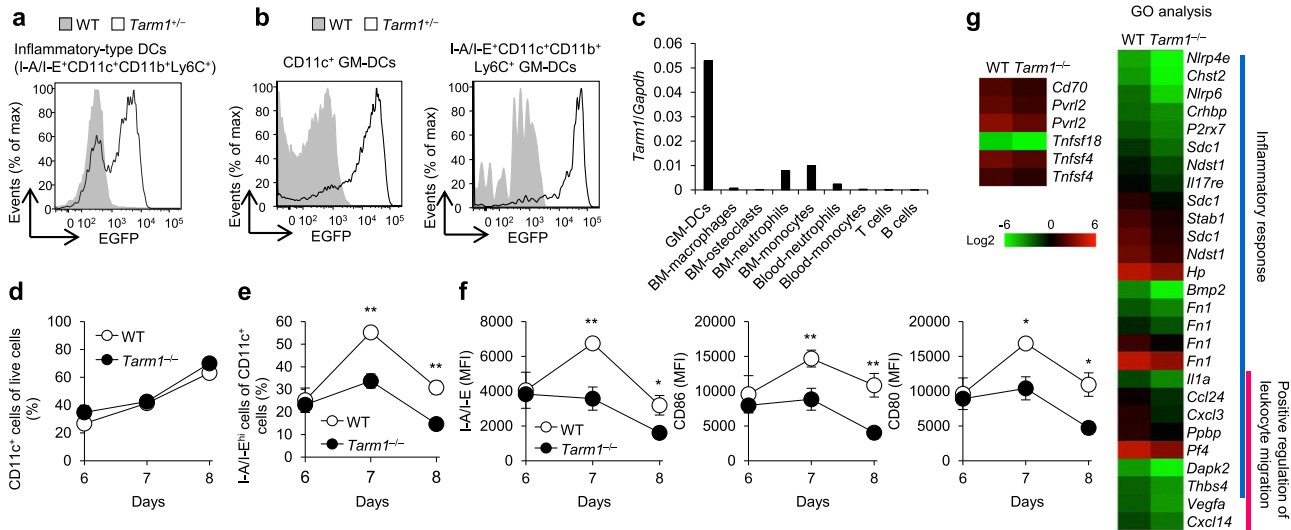

**Fig. 2 TARM1 is expressed by and is required for the activation of DCs. a** EGFP expression was examined in inflammatory-type DCs (I-A/I-E⁺CD11c⁺ CD11b⁺Ly6C⁺) of dLNs from WT and *Tarm1⁺/⁻* mice at day 7 after CIA induction by flow cytometry. Data are representative of two independent experiments. **b** EGFP expression was examined in GM-DCs from *Tarm1⁺/⁻* EGFP reporter mice by flow cytometry at day 8 after treatment with GM-CSF. Non-Tg WT mice were used as controls. Data are representative of three independent experiments. **c** *Tarm1* expression was examined in GM-DCs, BM macrophages, BM osteoclasts, BM neutrophils, BM monocytes, blood neutrophils, blood monocytes, T cells, and B cells from non-immunized WT mice using qPCR. Data are shown as mean of duplicate wells from a mouse and are representative of two independent experiments. **d, e** GM-DC differentiation from BM cells was examined in vitro. The proportion of CD11c⁺ (**d**) and I-A/I-E^hiCD11c⁺ cells (**e**) were analyzed in WT and *Tarm1⁻/⁻* GM-DCs by flow cytometry at the indicated days after treatment with GM-CSF. Data are representative of three independent experiments. Mean ± SD of triplicate wells. \*\**P* < 0.01 (two-tailed unpaired *t* test). **f** Expression of DC activation markers, I-A/I-E, CD86, and CD80, were examined in WT and *Tarm1⁻/⁻* GM-DCs after treatment with GM-CSF. Data are representative of three independent experiments. Mean ± SD of triplicate wells. \**P* < 0.05; \*\**P* < 0.01 (two-tailed unpaired *t* test). **g** Gene expression levels in WT and *Tarm1⁻/⁻* GM-DCs at day 8 after treatment with GM-CSF were analyzed by DNA microarray. Heat map shows log2 intensity of representative differentially expressed genes between WT and *Tarm1⁻/⁻* GM-DCs (fold change >2.0). Source data are provided as a Source data file and Supplementary Data 1.

GMDCs (Fig. 2g and Supplementary Data 1). Also, Gene ontology (GO) hierarchy analysis demonstrated that gene expression levels involved in inflammatory responses and leukocyte migration were impaired in *Tarm1⁻/⁻* GM-DCs. These observations indicate that TARM1 is required for the activation and maturation of DCs.

**Antigen-presenting ability of DCs is impaired in *Tarm1⁻/⁻* mice.** To understand the mechanism as to why CIA development is suppressed in *Tarm1⁻/⁻* mice, we investigated the recall T cell proliferation of dLN cells from WT and *Tarm1⁻/⁻* mice after CIA induction. The proliferative response of *Tarm1⁻/⁻* dLN cells to IIC was suppressed compared with that of WT dLN cells (Fig. 3a). The concentrations of proinflammatory cytokines, such as TNF, IFN-γ, and IL-17A, in the culture supernatant from *Tarm1⁻/⁻* LN cell culture were lower than those of WT LN cell culture (Fig. 3b–d).

Accordingly, we next examined DC maturation in dLNs after CIA induction. The proportion of CD11c⁺ cells in total LN cells was decreased only slightly in *Tarm1⁻/⁻* mice (Fig. 3e). However, we found that the proportion of I-A/I-E⁺CD11c⁺ mature DCs in total CD11c⁺ DCs was significantly lower in *Tarm1⁻/⁻* mice than in WT mice (Fig. 3e). Furthermore, I-A/I-E expression levels on CD11c⁺ cells were decreased in *Tarm1⁻/⁻*cells compared with WT counterparts (Fig. 3f).

Then we examined the antigen-presenting ability of DCs. DCs and T cells were highly purified from mice after IIC immunization and examined T cell-promoting activity of DCs against IIC. Both T cells from WT and *Tarm1⁻/⁻* mice were efficiently proliferated in response to IIC upon co-culture with WT DCs, but T cell responses against IIC were greatly reduced in the co-culture

with *Tarm1⁻/⁻* DCs (Fig. 3g). These results suggest that antigen-presenting ability of *Tarm1⁻/⁻* DCs is impaired.

We then assessed antigen incorporating ability of *Tarm1⁻/⁻* GM-DCs and found that the efficiency of AlexaFluor647 (AF647)-conjugated ovalbumin (OVA) incorporation by WT and *Tarm1⁻/⁻* GM-DCs was almost the same in terms of the AF647 intensity and positive cell percentage in flow cytometric analysis (Supplementary Fig. 2f). When WT and *Tarm1⁻/⁻* GM-DCs were co-cultured with carboxyfluorescein succinimidyl ester (CFSE)-labeled CD4⁺ T cells from OT-II mice, T cell proliferative response to OVA was decreased in cultures with *Tarm1⁻/⁻* GM-DCs compared with WT GM-DCs (Fig. 3h). Furthermore, in allogeneic mixed lymphoid reaction, proliferation of CFSE-labeled T cells from BALB/cA mice was decreased in co-cultures with C57BL/6J-*Tarm1⁻/⁻* GM-DCs (Fig. 3i). However, proliferative responses of T and B cells were similar between *Tarm1⁻/⁻* and WT cells upon stimulation with anti-CD3 and anti-IgM antibodies, respectively (Supplementary Fig. 2g, h), and differentiation of *Tarm1⁻/⁻* CD4⁺ T cells to regulatory T, T helper type 1 (Th1), and Th17 cells was normal (Supplementary Fig. 2i, j), indicating that intrinsic T and B cell functions are normal in *Tarm1⁻/⁻* mice. These results clearly indicate that antigen-presenting ability of GM-DCs from *Tarm1⁻/⁻* BM cells is impaired compared with WT GM-DCs.

**IIC is an endogenous ligand for TARM1 and controls the maturation of GM-DCs in a TARM1-dependent manner.** Next, we analyzed the ligand for TARM1. Presence of endogenous ligand(s) for TARM1 in BM-DCs was suggested, because the effect of TARM1 deficiency was observed in the differentiation of GM-DCs from BM cells. We therefore searched for the TARM1 ligand on GM-DCs by flow cytometric analysis using TARM1-Fc,

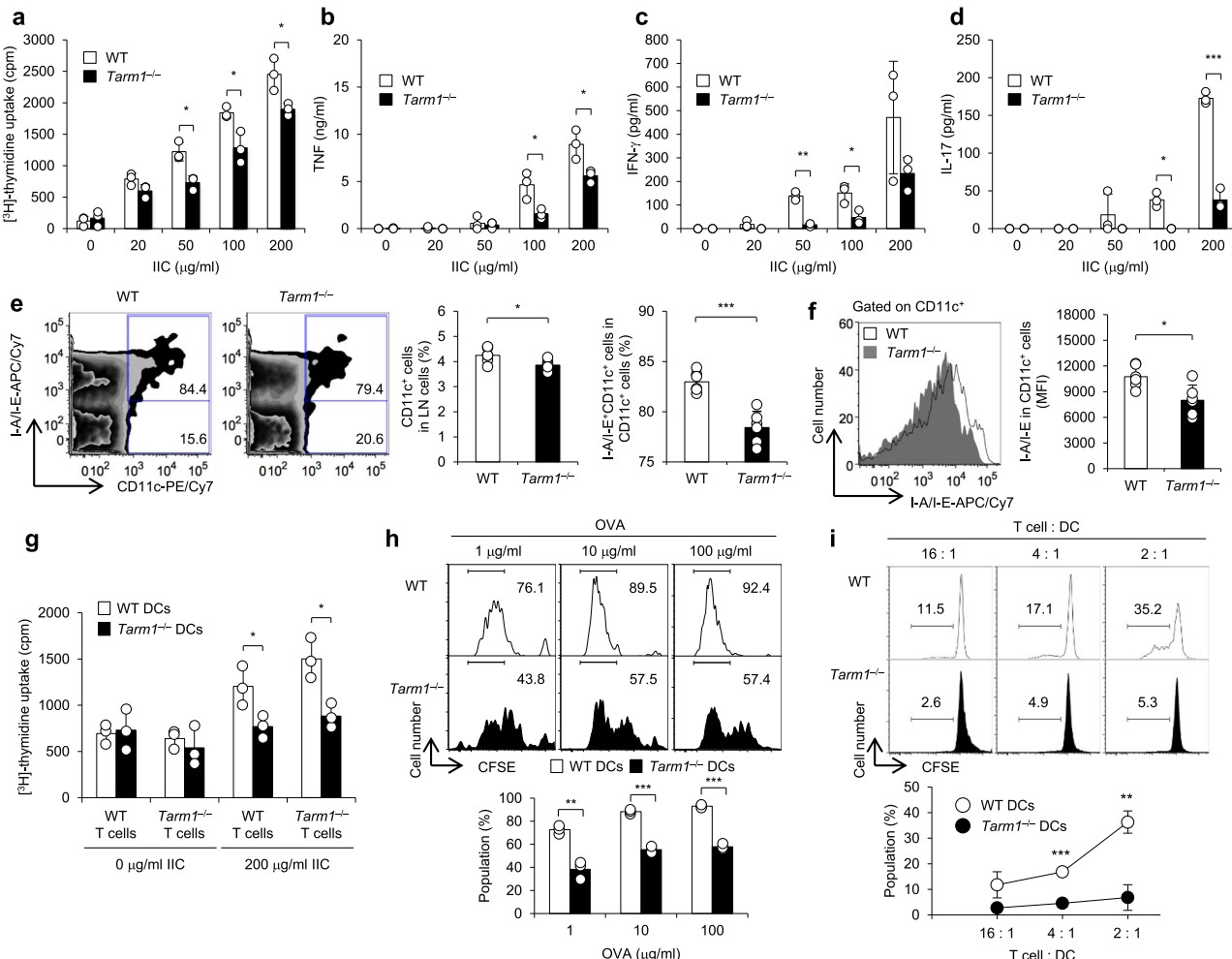

**Fig. 3 Antigen-presenting ability of DCs is impaired in *Tarm1−/−* mice. a** At day 10 after immunization, LN cells from WT and *Tarm1−/−* mice were re-stimulated with the indicated concentrations of IIC for 66 h. [³H]-thymidine incorporation into acid-insoluble fraction was determined. Data are representative of three independent experiments. Mean ± SD *P < 0.05 (two-tailed unpaired *t* test). **b–d** Cytokine concentrations in culture supernatants in **a** were determined by ELISA. Data are representative of three independent experiments. Mean ± SD. *P < 0.05; **P < 0.01; ***P < 0.001 (two-tailed unpaired *t* test). **e** The proportions of CD11c⁺ cells in inguinal LN cells and I-A/I-E⁺CD11c⁺ cells in CD11c⁺ cells from WT and *Tarm1−/−* mice were analyzed by flow cytometry at day 10 after immunization. Data are representative of three independent experiments. WT = 6, *Tarm1−/−* = 6. Mean ± SD. *P < 0.05; ***P < 0.001 (two-tailed unpaired *t* test). **f** The expression of I-A/I-E on CD11c⁺ LN cells from WT and *Tarm1−/−* mice were analyzed by flow cytometry at day 10 after immunization. Data are representative of three independent experiments. WT = 6, *Tarm1−/−* = 6. Mean ± SD. *P < 0.05 (two-tailed unpaired *t* test). **g** At day 10 after immunization, DCs and T cells from WT and *Tarm1−/−* mouse dLNs were co-cultured in the presence of IIC. T cell proliferation was determined by [³H]-thymidine incorporation. Data are representative of three independent experiments. Mean ± SD. *P < 0.05 (two-tailed unpaired *t* test). **h** CFSE-labeled OT-II CD4⁺ T cells were co-cultured with WT and *Tarm1−/−* GM-DCs in the presence of the indicated concentrations of OVA. Three days later, CFSE intensity was assessed by flow cytometry. Data are representative of three independent experiments. Mean ± SD of triplicate wells. **P < 0.01; ***P < 0.001 (two-tailed unpaired *t* test). **i** T cells from BALB/cA mice were co-cultured with WT and *Tarm1−/−* GM-DCs from C57BL/6J mice. Five days later, CFSE intensity was assessed by flow cytometry. Data are representative of two independent experiments. Mean ± SD of triplicate wells. **P < 0.01; ***P < 0.001 (two-tailed unpaired *t* test). Source data are provided as a Source data file.

a soluble form of TARM1 ectodomain fused with IgG Fc portion, as the probe. TARM1-Fc selectively bound to CD11c⁺ cells but not to CD11c⁻ cells (Fig. 4a). Because many of Ig-like family members can self-ligate in a *cis*-manner[2,3], we further examined TARM1-Fc binding to *Tarm1−/−* GM-DCs to exclude the possibility that TARM1 bound another TARM1 on the cell surface. We found that TARM1-Fc also bound to *Tarm1−/−* CD11c⁺ cells, indicating that the binding was not mediated by TARM1 (Supplementary Fig. 3a). We also found that TARM1-Fc bound to CD11b⁺CD11c⁺ and F4/80⁺CD11b⁺ cells from LNs and spleens of WT mice (Supplementary Fig. 3b, c). However, no TARM1-Fc binding was observed to CD4⁺ T and CD8⁺ T and B cells of LNs and spleens from WT mice under physiological conditions (Supplementary Fig. 3b, c) and after CIA induction

(Supplementary Fig. 3d, e), indicating that these cells do not express the ligand. Also, we observed no binding of TARM1-Fc to in vitro differentiated Th1 and Th17 cells (Supplementary Fig. 3f).

It was reported that collagens are functional ligands for OSCAR[12,32], and their interaction is involved in DC maturation[33]. In addition, collagenous molecules are recognized by other LILR members such as leukocyte-associated Ig-like receptor-1 (LAIR-1) and glycoprotein VI (GPVI)[34–36]. Thus we examined the possibility that collagenous molecules are endogenous ligands for TARM1. In solid-phase binding assay, TARM1-Fc directly bound to type I collagen (IC) and IIC, but not to bovine serum albumin (BSA), in a concentration-dependent manner (Fig. 4b). We then examined the expression of *Col1a1* and *Col2a1* in

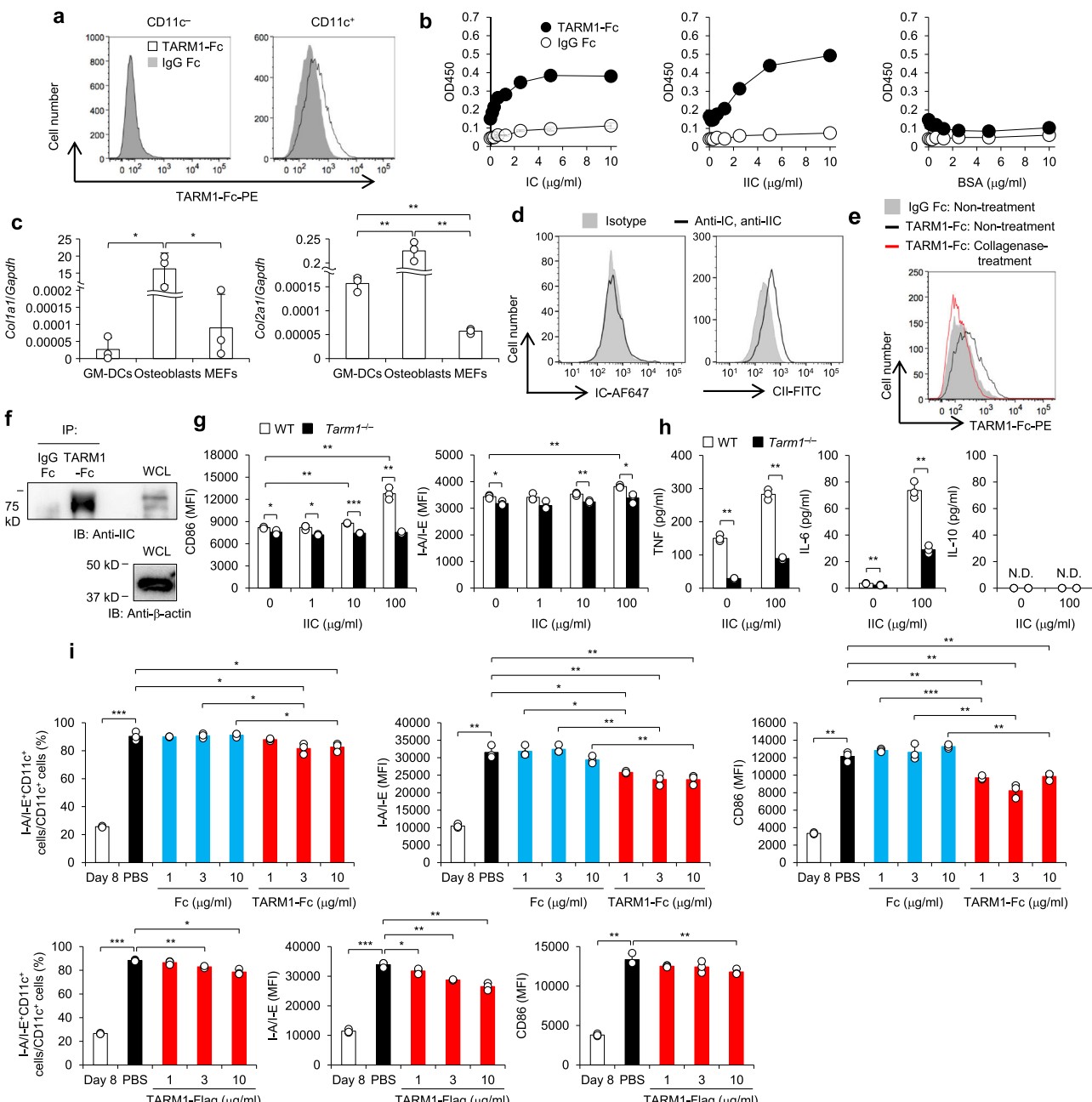

**Fig. 4 IIC is a functional ligand for TARM1 and controls GM-DC maturation in a TARM1-dependent manner. a** Bindings of TARM1-Fc to WT GM-DCs were analyzed by flow cytometry at day 8 after GM-CSF treatment. IgG Fc was used as a negative control. **b** Bindings of TARM1-Fc and IgG Fc to plate-coated IC, IIC, and BSA were analyzed by solid-phase binding assay. **c** Expression levels of *Col1a1* and *Col2a1* in GM-DCs, osteoblasts, and MEFs were analyzed by qPCR. **d** Cell surface expression of IC and IIC on GM-DCs at day 8 was analyzed by flow cytometry using specific antibodies. **e** Binding of TARM1-Fc to non- or collagenase-treated GM-DCs was analyzed by flow cytometry. IgG Fc was used as a negative control. **f** Whole GM-DC lysate was incubated with TARM1-Fc- and IgG Fc-beads, and after collection of beads, proteins were recovered. Then these proteins were immunoblotted with anti-IIC antibodies. Whole-cell lysate (WCL) was also immunoblotted with anti-IIC and anti-β-actin antibodies. **g** Expression of DC activation markers (CD86 and I-A/I-E) on WT and *Tarm1*⁻/⁻ GM-DCs was analyzed by flow cytometry after stimulation with different concentration of soluble IIC for 24 h. **h** Cytokine concentrations (TNF, IL-6, and IL-10) of culture supernatants in **g** were determined by flow cytometry with cytometric beads. **i** GM-DCs at day 8 after GM-CSF treatment were stimulated with plate-bound IIC (1 μg/ml) for 24 h in the absence (PBS) or presence of TARM1-Fc, Fc, or TARM1-Flag (1, 3, and 10 μg/ml). The expression levels of I-A/I-E and CD86 in CD11c⁺ cells before and after IIC stimulation were analyzed by flow cytometry. The expression levels of these molecules on GM-DCs at day 8 before IIC treatment were shown as Day 8. All the data are representative of three independent experiments, except two experiments in **i**. Mean ± SD of triplicate wells. N.D. not detected. *P < 0.05; **P < 0.01; ***P < 0.001 (two-tailed unpaired *t* test). Source data are provided as a Source data file.

GM-DCs by qPCR and found that *Col2a1* is expressed in GM-DCs, although the expression levels were very low compared with osteoblasts (Fig. 4c). We also detected IIC expression on GM-DCs by flow cytometry (Fig. 4d). Collagenase treatment canceled

TARM1-Fc binding to GM-DCs, suggesting that collagenous molecules are involved in the binding (Fig. 4e). We also found that TARM1-Fc formed a complex with IIC in GM-DC lysates by employing immunoprecipitation–western blotting assay (Fig. 4f).

We then stimulated GM-DCs with IIC and analyzed the expression of DC activation makers by flow cytometry. Expression of CD86 and I-A/I-E in $Tarm1^{-/-}$ GM-DCs was significantly low compared with those of WT GM-DCs and enhanced by the addition of IIC in WT, but not in $Tarm1^{-/-}$ GM-DCs (Fig. 4g). TNF and IL-6 production from GM-DC cultures were also significantly low in $Tarm1^{-/-}$ GM-DCs and increased by the treatment with IIC in a TARM1-dependent manner (Fig. 4h). Upon stimulation with LPS, CpG, poly(I:C), zymosan, and *Mycobacterium tuberculosis*, productions of TNF, IL-6 and IL-10 were similar between WT and $Tarm1^{-/-}$ GM-DCs, indicating that intrinsic functions of $Tarm1^{-/-}$ GM-DCs are normal (Supplementary Fig. 3g).

Next, we treated GM-DCs with plate-bound IIC and examined the effect on I-A/I-E and CD86 expression. As shown in Fig. 4i, plate-bound IIC greatly enhanced maturation marker expression on CD11c$^+$ cells. Then we analyzed the effect of soluble TARM1-Fc on IIC-induced GM-DC activation. We found that the content of I-A/I-E$^+$CD11c$^+$ subpopulation in CD11c$^+$ population as well as the expression level of I-A/I-E and CD86 on CD11c$^+$ cells was decreased by the treatment with TARM1-Fc compared to control IgG Fc (Fig. 4i). Furthermore, we found that soluble TARM1-Flag treatment also decreased the I-A/I-E$^+$CD11c$^+$ population in CD11c$^+$ cells and the expression levels of I-A/I-E and CD86 on CD11c$^+$ cells. These results indicate that plate-bound IIC greatly enhance GM-CSF maturation and soluble TARM1 can suppress the maturation, and the TARM1 portion, but not the Fc portion, of the TARM1-Fc molecule is responsible for the suppression.

Because $Tarm1$ is weakly expressed in BM neutrophils, we examined the roles of TARM1 in BM neutrophils. We analyzed cytokine production in WT and $Tarm1^{-/-}$ BM neutrophils after stimulation with IIC, LPS, zymosan, CpG, poly(I:C), and *M. tuberculosis* (Supplementary Fig. 3h). We found that TNF production was reduced in $Tarm1^{-/-}$ BM neutrophils upon stimulation with zymosan and *M. tuberculosis*, and IL-6 production was decreased following stimulation with LPS and zymosan, but we could not detect TNF, IL-6, and IL-10 production in IIC-treated cells. TNF and IL-6 production were rather enhanced after CpG treatment in $Tarm1^{-/-}$ BM-neutrophils compared with WT neutrophils.

**Administration of TARM1-Fc attenuates CIA.** Finally, we examined the therapeutic effect of TARM1-Fc on CIA. We injected TARM1-Fc into the articular cavity of an arthritic knee joint of a CIA-induced DBA/1J mouse and IgG Fc into another knee joint of the same mouse. Administration of TARM1-Fc resulted in significant reduction of arthritic score of the ankle joint, but not of the IgG Fc-injected foot (Fig. 5a, b). Histological examination revealed that synovial inflammation, pannus formation, and cartilage and bone destruction in the joints were suppressed in TARM1-Fc-treated foot compared to IgG Fc-treated foot of the same mouse (Fig. 5c, d). In the joints of TARM1-Fc-treated foot, *Il10* expression was enhanced. Furthermore, the expression of *Myeloperoxidase* (*Mpo*) and *Ccl5* was reduced, indicating that neutrophil recruitment was suppressed in TARM1-Fc-treated mice (Fig. 5e). These results suggest that TARM1 inhibition is effective to attenuate autoimmune arthritis.

## Discussion

In this report, we showed that the development of CIA is greatly suppressed in $Tarm1^{-/-}$ mice. Serum IIC-specific antibody levels and IIC-specific T cell recall proliferative responses were reduced in $Tarm1^{-/-}$ mice, because antigen-presenting ability of DCs is impaired in $Tarm1^{-/-}$ mice. Furthermore, we showed that soluble TARM1-Fc can attenuate development of CIA. *Ccl5* expression

and recruitment of neutrophils to joints were suppressed in these TARM1-Fc-treated mice. Thus TARM1 plays an important role in the development of CIA by promoting maturation/activation of DCs.

We found that $Tarm1$ is prominently expressed in GM-DCs, in consistent with a previous report[7]. However, the expression in BM neutrophils, blood neutrophils, BM monocytes, and blood monocytes was relatively low. When $Tarm1$ expression was monitored using $Tarm1^{+/-}$ EGFP reporter mice, $Tarm1$ expression in DCs was low under physiological conditions, but after CIA induction, $Tarm1$ expression was greatly increased in I-A/I-E$^{hi}$Ly6C$^+$CD11b$^+$CD11c$^+$ inflammatory-type DCs. When DCs were induced to differentiate from BM cells in vitro, $Tarm1$ was highly expressed in GM-DCs, especially in an inflammatory-type subset of GM-DCs, and weakly in CD11b$^+$ FL-DCs but not in CD24$^+$ or B220$^+$ FL-DCs. $Tarm1$ expression in GM-DCs was further increased following stimulation with inflammatory cytokines and PAMPs as reported by a previous report[7].

When $Tarm1^{-/-}$ BM cells were induced to differentiate with GM-CSF, DC maturation, as monitored by the expression of I-A/I-E, CD80, and CD86, was impaired. When $Tarm1^{-/-}$ GM-DCs were used as antigen-presenting cells, OVA-specific proliferative T cell response as well as mixed lymphocyte culture T cell proliferation was greatly reduced compared with WT DCs, indicating impairment of antigen-presenting ability of $Tarm1^{-/-}$ GM-DCs. Because DC maturation is required for antigen presentation[37], these results suggest that TARM1 signaling is crucial for GM-DC maturation.

Consistent with this notion, the proportion of I-A/I-E$^+$ mature DCs was decreased in $Tarm1^{-/-}$ mice after induction of CIA. Recall that T cell proliferative response and cytokine production upon IIC treatment of LN cells were significantly reduced in $Tarm1^{-/-}$ mice. Furthermore, $Tarm1^{-/-}$ DCs failed to support IIC-specific proliferation of T cells from CIA-induced WT and $Tarm1^{-/-}$ mice. Thus TARM1 signaling plays an important role in the development of CIA by promoting antigen-presenting ability of DCs. The decrease of I-A/I-E$^+$ mature DC population in $Tarm1^{-/-}$ mice (Fig. 3e, f) was relatively small as compared with great reduction of the antigen-presenting ability and T cell-stimulating activity (Fig. 3a–d). We think this is because I-A/I-E and CD86 represent only a part of molecules that are involved in antigen presentation. As shown in Fig. 2g, the expression of many genes was decreased in $Tarm1^{-/-}$ mice, and probably many of these genes are coordinately involved in antigen presentation in a synergistic or additive manner.

We showed that IIC is a functional ligand for TARM1. Some of LILR family members, such as OSCAR, LAIR-1, LAIR-2, and GPVI, recognize collagens and regulate cellular functions, such as osteoclast formation and DC maturation[12,33–36,38]. In this study, we found that TARM1 also directly binds to IIC and IC and forms a complex with IIC. Upon stimulation with IIC, the expression of DC maturation markers such as I-A/I-E and CD86 and the production of inflammatory cytokines such as TNF and IL-6 was enhanced in WT GM-DCs but not in $Tarm1^{-/-}$ GM-DCs. Plate-bound IIC showed much efficient maturation-enhancing activity compared with soluble IIC (Fig. 4g, i), suggesting that topological distribution of TARM1 may affect the signaling.

We found that activated DC-derived inflammatory cytokines enhance $Tarm1$ expression in GM-DCs (Fig. 4h and Supplementary Fig. 2c). Actually, the expression of $Tarm1$ was enhanced in the affected joints of RA model mice. Furthermore, gene expression analysis revealed that the expression of a group of inflammation-related genes is reduced in $Tarm1^{-/-}$ GM-DCs. Thus these observations indicate that $Tarm1$ expression in DCs is enhanced by the cytokines produced by DCs, suggesting that

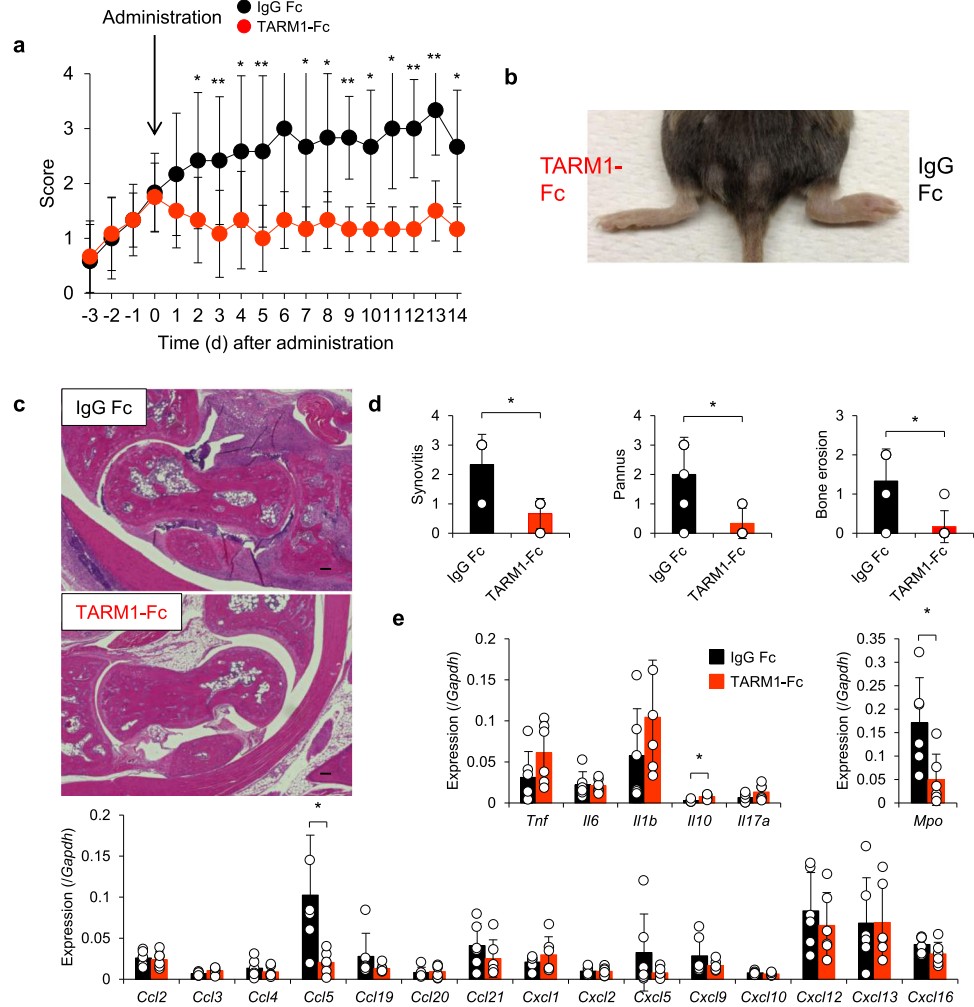

**Fig. 5 Administration of TARM1-Fc attenuates the development of CIA. a** Severity score of arthritic joints from CIA-induced DBA/1J mice treated with TARM1-Fc or IgG Fc. TARM1-Fc and IgG Fc (1 µg/30 µl for each) were administrated into the articular cavity of the left and right knee joints of a CIA-induced mouse, respectively, and the severity scores of IgG Fc- (black) or TARM1-Fc-injected paw (red) were measured separately. Data from two independent experiments are combined ($n = 12$). Mean ± SD. *$P < 0.05$; **$P < 0.01$ (two-tailed Mann–Whitney $U$ test). **b** Representative photograph of feet treated with TARM1-Fc (left) or IgG Fc (right) as shown in **a**. **c**, **d** Histology of the HE-stained ankle joint at day 14 after TARM1-Fc or IgG Fc treatment (**c**). Data are representative of six mice. Histological score of synovitis, pannus formation, and bone erosion of ankle joints are shown in (**d**). Mean ± SD. Scale bar, 100 µm. *$P < 0.05$ (two-tailed Mann–Whitney $U$ test). **e** Gene expressions in arthritic ankles and tarsal joints from TARM1-Fc- or IgG Fc-treated legs were analyzed by qPCR ($n = 6$). Mean ± SD. *$P < 0.05$ (two-tailed unpaired $t$ test). Source data are provided as a Source data file.

TARM1 promotes DC maturation through interaction with IIC on DCs by forming a self-amplification loop. Because IIC is also detected on CD11b$^+$ DCs and macrophages in LNs and spleen, IIC and TARM1 interaction may occur in both *cis* and *trans* manners.

IIC is an extracellular matrix component of articular cartilage and is one of autoantigens in RA[39,40]. Thus it is conceivable that TARM1 is required for the internalization and antigen presentation of IIC by forming a complex with TARM1 specifically in CIA. However, this seems unlikely, since TARM1 deficiency in DCs also affected IIC-independent T cell responses, such as OVA-specific T cell response and T cell response in mixed lymphocyte cultures. Thus TARM1 signaling enhances not only presentation of IIC but also presentation of other antigens by DCs. Because IIC is rich in joints, DC maturation may be preferentially induced in joints through activation of TARM1. This may partially explain why arthritis preferentially develops in autoimmune-prone mice[25,26,41].

Although antigen-presenting ability of *Tarm1*$^{-/-}$ DCs was greatly reduced, phagocytic incorporation of OVA and cytokine production upon various innate immune stimulations, such as LPS, CpG, poly(I:C), zymosan and Mycobacteria, were not affected, indicating that the intrinsic DC functions other than antigen-presenting ability is normal in *Tarm1*$^{-/-}$ DCs. Cytokine production was reduced in *Tarm1*$^{-/-}$ BM neutrophils following stimulation with zymosan, *Mycobacteria*, and LPS, suggesting some functional defects in neutrophils. Thus it is possible that these defects of neutrophils are also involved in the defect of CIA development. However, the defects in neutrophils cannot explain completely the defects observed in *Tarm1*$^{-/-}$ mice, because neutrophils are not involved in the T cell sensitization and antibody production. Given that *Tarm1* is most highly expressed in GM-DCs among BM-derived cells (Fig. 2c), and TARM1 controls antigen presentation and cytokine production of DCs upon stimulation with IIC (Fig. 4g, h), the defects in DCs are considered to be the main reason for the defect of CIA development in *Tarm1*$^{-/-}$ mice. Thus reduced MPO expression in CIA-induced joints after treatment with TARM1-Fc is considered to be a result of reduced neutrophil infiltration due to suppression of chemokine expression and neutrophil function.

We showed that treatment with soluble TARM1 can suppress IIC-induced activation of GM-DCs. In a previous study, Radjabova et al. showed that addition of TARM1-Fc to a $CD4^+$ T cell culture can suppress the activation/proliferation of T cells and suggested that endogenous ligand(s) of TARM1 is expressed on T cells and transduces signal to inhibit T cell activation and proliferation[7]. However, we could not observe any effect of TARM1 deficiency on T/B cell proliferation and Th1/Th17 differentiation. We do not know the reason for this apparent discrepancy at present and clearly further studies are needed to explain these results.

By sharing a common ligand, paired Ig-like receptors, which transduce opposing signals through ITAM and immunoreceptor tyrosine-based inhibition motif (ITIM), balance the host immune responses[42,43]. LAIR-1, a member of the ITIM-bearing LILR family, is expressed on various types of immune cells including DCs[44] and suppresses cell activation by binding collagens[45]. In patients with RA, LAIR-1 expression is closely associated with the pathogenesis[46]. Recently, Kim et al. demonstrated that CIA is suppressed by LAIR-1 signaling in mice[47]. Because LAIR-1 and TARM1 share collagens as common ligands, LAIR-1 and TARM1 may function as paired receptors to regulate DC activation. This explains well why CIA is greatly suppressed in $Tarm1^{-/-}$ mice, because only inhibitory LAIR-1 receptor is functional in these mice.

Taken together, we have shown that TARM1 plays an important role for the maturation and activation of DCs through interaction with IIC (Supplementary Fig. 4). Because excess DC activation is suggested in many autoimmune and allergic diseases, and possible associations of single-nucleotide polymorphisms of $TARM1$ with RA and uveitis are suggested in published data (rs654765, rs112143130, rs73060651, rs112227086 and rs61416627; $P < 0.05$)[48], our observations suggest that TARM1 is a good target for the development of new drugs to treat such diseases.

## Methods

**Mice.** C57BL/6J, BALB/cA and DBA/1J mice were purchased from Japan SLC Inc. (Tokyo, Japan) or CLEA Japan (Tokyo, Japan). OT-II TCR Tg mice were gifts from H. Kishimoto at RIBS, Tokyo University of Science. All mice were maintained under SPF conditions in environmentally controlled clean rooms at RIBS, Tokyo University of Science and Chiba University. Age-matched (6–12 weeks old) and sex-matched mice were used for all experiments. The experiments were carried out according to the institutional ethical guidelines for animal experiments and the safety guidelines for gene manipulation experiments and were approved by the institutional committees.

**Generation of $Tarm1$-deficient mice.** Genomic DNA was isolated from EGR-101 ES cells from C57BL6 embryos[49]. DNA fragments for 5′ and 3′ arms were amplified by PCR with the genomic DNA using the primer sets (Supplementary Table 1). The lengths of the 5′ and 3′ arms are 3.56 and 4.92 kbp, respectively. The targeting vector was constructed by replacing the genomic locus containing the exon 1 of the $Tarm1$ gene, which contained the initiation codon, by an EGFP gene and a neomycin resistance gene ($Neo^r$) under the control of a PGK1 promoter. A diphtheria toxin A gene under the MC1 promoter was ligated to the 3′ end of the targeting vector. The targeting vector was linearized by $NotI$ treatment, and the digested vector was electroporated into C57BL/6 mouse ES cells[49]. ES cell clones were selected in the presence of G418 (Nacalai Tesque) and screened by PCR with the primer pair (Supplementary Table 1). Homologous recombination was confirmed by Southern blot analysis using 5′ and 3′ probes. The 5′ and 3′ probes were amplified by PCR with the primer sets (Supplementary Table 1). Chimeric mice were generated by an aggregation method with a clone of targeted ES cells. To generate heterozygous offspring, male chimeric mice were mated with C57BL/6 female mice. Disruption of allele targeting $Tarm1$ was confirmed by PCR with the primer pair (Supplementary Table 1). The common primer and the WT primer were used to detect the WT allele (approx. 300 bp), and the common primer and mutant primer were used to detect the mutant allele (approx. 400 bp). $Tarm1^{-/-}$ mice were normally born at the expected Mendelian ratio.

**Collagen-induced arthritis.** Complete Freund's adjuvant (CFA) was prepared by mixing heat-killed $M. tuberculosis$ H37 Ra (1 mg/ml; Difco, Detroit, MI) in

incomplete Freund's adjuvant (ThermoFisher SCIENTIFIC, Waltham, MA). An emulsion was formed by dissolving 4 mg/ml chicken IIC (SIGMA-ALDRICH, St. Louis, MO) with the CFA (vol:vol = 1:1). Mice (8–10 weeks old) were immunized by intradermal injection with 200 μl of the emulsion at three sites near the base of tail. Fourteen days after first immunization, mice were given booster injection with the same amount of the emulsion. The swelling of each paw was graded as follows: score 0, no swelling; score 1, swelling or focal redness of finger joints; score 2, mild swelling of wrist or ankle joints; score 3, severe swelling of the entire paw. The score of four paws were totaled, i.e., maximal score = 12.

For examination of the therapeutic effect of TARM1-Fc on arthritis, we injected TARM1-Fc (1 μg/30 μl/knee joint) or IgG Fc (1 μg/30 μl/knee joint) into the articular cavity of the left or right knee joint of CIA-induced DBA1/J mice with mild score[30]. DBA1/J mice were immunized with 100 μl of an emulsion consisting of 2 mg/ml IIC and 1.65 mg/ml CFA on days 0 and 21[30]. Arthritis development was graded in TARM-1-Fc- and IgG Fc-injected paw separately as follows: score 0, no change; score 1, erythema and mild swelling confined to the tarsal joints; score 2, erythema and mild swelling extending from the tarsal joint to digits; score 3, erythema and moderate swelling extending from metatarsal joints; score 4, erythema and severe swelling encompassing the ankle, foot, and digits, or ankylosis of the limb. Maximal scores = 4.

**Histopathology.** The paraffin-embedded section preparation and immunohistochemical staining were conducted by the Division of Molecular Pathology, The Institute of Medical Science, The University of Tokyo. Ankle joints on day 50 of CIA were fixed with 10% neutral-buffered formalin at 4 °C overnight and decalcified with 10% EDTA/PBS for 2–3 weeks. They were embedded in paraffin and slices were prepared. Serial tissue sections (5 μm) were stained with HE or Safranin O. Histological sections were graded on a scale of 0–3: score 0, no sign of inflammation; score 1, mild inflammation with hyperplasia of the synovial lining without cartilage destruction; score 2, grade 1 changes plus granulomatous lesions in the synovial sublining tissue; score 3, grade 2 changes plus pannus formation and cartilage/bone destruction[50]. Arthritis index of the ankle joint was estimated from the grade of talus and bones including tibia and calcaneum of each mouse.

**Antibody titers.** Sera were collected at day 42 after first immunization. Chicken IIC (20 μg/ml) was coated to 96-well plates. Serially diluted serum samples were incubated at room temperature for 1 h. After washing with PBS/0.05% Tween20, wells were incubated with alkaline phosphatase-conjugated rabbit anti-mouse IgG (1:1000 dilution; Zymed, San Francisco, CA; cat# 62-6622; lot# 20369646) at room temperature for 1 h. Substrate phosphatase SIGMA104 (SIGMA-ALDRICH) was added, and the absorbance at 415 nm was measured using a microplate reader, MTP-300 analyzer (Hitachi, Tokyo, Japan).

**Flow cytometry.** Antibodies used for flow cytometric analysis are listed in Supplementary Table 2. Cells were treated with 1 μg/ml anti-CD16/32 antibody (clone 2.4G2)/FACS buffer (HBSS containing 2% fetal bovine serum (FBS) and 0.01% sodium azide) at 4 °C for 20 min. The cells were incubated with fluorescent-conjugated antibodies on ice for 30 min (Supplementary Table 2) or with 10 μg/ml Fc-fused proteins on ice for 1 h followed by 5 μg/ml fluorescent-conjugated anti-human IgG on ice for 1 h. In the binding assay, cells were treated with 200 U/ml collagenase (Wako Chemicals, Tokyo, Japan) at 37 °C for 30 min. For staining of collagens on GM-DCs, anti-IC (8D4A1; 1 and 5 μg/ml; Chondrex, Redmond, WA; cat# 7041; lot# 120442) and anti-IIC antibodies (2B1.5, 1 and 5 μg/ml; ThermoFisher SCIENTIFIC; cat# MA5-12789; lot# VA2926216A) were used. Cells were washed, suspended, filtered through a 70-μm filter, and analyzed by flow cytometers, FACSCantoII or FACSVerse (BD Biosciences, Sparks, MD). Data analysis was performed using FACSDiva (Version 6.1.2), FACSuite (Version 1.0.5.3841), and FlowJo (Version 9 and 10; Tree Star, Ashland, OR). Dead cells were stained with 2 μg/ml 7-amino-actinomycin D (SIGMA-ALDRICH) and 10 μg/ml propidium iodide (SIGMA-ALDRICH). Gating strategies for the characterization of cell populations are provided in Supplementary Figs. 5 and 6. Application of anti-collagen antibodies to flow cytometry was tested using collagen-producing osteoblasts (Supplementary Fig. 6f). For intracellular staining, cells were cultured in the presence of 50 ng/ml phorbol myristate acetate, 500 ng/ml ionomycin, and 2 μM monensin for 5 h. After blocking with anti-CD16/32 antibody, cells were incubated with fluorescence-conjugated antibodies. Cells were washed with FACS buffer, fixed with 4% paraformaldehyde on ice for 20 min, and permeabilized with 0.1% saponin/FACS buffer. After a wash with 0.1% saponin/FACS buffer, cells were further incubated with fluorescence-antibodies for intracellular cytokine staining on ice for 30 min. For intracellular Foxp3 staining, Fixation/Permeabilization Concentrate and Diluent solutions (eBioscience) were used in accordance with the manufacturer's instruction. For cell sorting, single-cell suspensions were incubated with 2 μg/ml biotin-conjugated anti-Ter119 (TER-119, BD Pharmingen, San Diego, CA) and 2 μg/ml biotin-conjugated anti-B220 antibodies (RA3-6B2, BioLegend) followed by anti-biotin beads (Miltenyi Biotech, Bergisch Gladbach, Germany). The labeled cells were isolated using a magnetic cell sorting system autoMACS (Miltenyi Biotech) in accordance with the manufacturer's instruction. The negative cells were stained with fluorescent antibody, and $CD11c^+$ cells and $CD4^+CD3^+$

cells were highly purified using flow cytometers, FACSAria II (Becton Dickinson) or MoFlo XDP (Beckman Coulter, Miami, FL).

**Reverse transcription and qPCR**. RNAs from tissues and cells were purified using the Sepasol-RNA I Super (Nacalai Tesque, Kyoto, Japan) and GenElute mammalian total RNA miniprep kit (SIGMA-ALDRICH), respectively. Resulting RNAs were reverse-transcribed using the high-capacity cDNA reverse transcription kit (Applied Biosystems, Foster City, CA). For qPCR analysis, SYBR Premix Ex Taq I or SYBR Premix Ex Taq II kits (Takara, Kyoto, Japan) were used with specific primer sets (Supplementary Table 3).

**Preparation of BM-derived myeloid cells**. BM was extracted from the tibia and femur, and red blood cells were destroyed using hemolysis buffer (140 mM $NH_4Cl$ and 17 mM Tris-HCl, pH 7.2). BM cells were cultured at $2 \times 10^5$ cells/ml RPMI1640 supplemented with 10% FBS, 100 U/ml penicillin, 100 μg/ml streptomycin, 1% 2-mercaptoethanol (R10), and recombinant mouse GM-CSF (20 ng/ml; Peprotech, Rocky Hill, NJ) in non-treated dishes. On day 3, the same volume of fresh R10 medium containing GM-CSF (20 ng/ml) was further added to the dishes. On day 6, approximately half of the medium was replaced with fresh R10 medium containing 10 ng/ml GM-CSF. Loosely adherent and non-adherent cells were collected on the indicated day and used for subsequent experiments. For a concentration-dependent experiment, different concentration of GM-CSF was used (on days 0 and 3; 2, 10, and 20 ng/ml: on days 6 and 8; 1, 5, and 10 ng/ml). For preparation of FL-DCs, BM cells were cultured at $2 \times 10^6$ cells/ml RPMI1640 supplemented with 10% FBS, 100 U/ml penicillin, 100 μg/ml streptomycin, 1% 2-mercaptoethanol, and recombinant mouse Flt3L (100 ng/ml; Peprotech) for 10 days. Adherent cells were collected by pipetting with 2.5 mM EDTA/PBS and used as FL-DCs. For preparation of BM macrophages, BM cells were cultured at $2 \times 10^6$ cells/ml RPMI1640 supplemented with 10% FBS, 100 U/ml penicillin, 100 μg/ml streptomycin, 1% 2-mercaptoethanol, and recombinant human M-CSF (20 ng/ml; R&D systems). On day 3, half volume of fresh medium containing 20 ng/ml M-CSF was further added to the dishes. On day 7, adherent cells were collected by scraping with 2.5 mM EDTA/PBS and used as BM macrophages.

**Preparation of BM-derived osteoclasts**. BM cells were preincubated in 100-mm dishes for >2 h. Non-adherent cells ($3 \times 10^5$/well in 96-well plates) were cultured in α-MEM supplemented with recombinant human M-CSF (20 ng/ml; R&D systems). Two days later, medium were replaced to M-CSF and human recombinant RANKL (20 and 100 ng/ml; Oriental Yeast Co., Tokyo, Japan). On day 4, the medium was further replaced with fresh medium supplemented with the cytokines. On day 5, cells were collected.

**Preparation of osteoblasts and mouse embryonic fibroblasts (MEFs)**. Primary osteoblasts were isolated from calvariae of new born mice. Calvariae were predigested with 0.1% collagenase (Wako Chemicals, Osaka, Japan) and 0.1% dispase (Roche Diagnostics, Mannheim, Germany) at 37 °C for 20 min. The primary supernatants were removed and calvariae were further treated with the collagenase/dispase solution for 1 h. Single-cell suspensions were prepared using a cell strainer (70 μm; BD Biosciences, San Diego, CA). For the preparation of MEFs, pregnant mice on day 13.5–15.5 were anesthetized with isoflurane and embryos were separated. After removal of the head and internal organs, the embryos were minced and treated with 0.1% trypsin/EDTA (Sigma-Aldrich) with gentle shaking for 15 min. Single-cell suspensions were prepared using a cell strainer (70 μm).

**Preparation of neutrophils, monocytes, T cells, and B cells**. Cells from BM, blood, spleens, or LNs were blocked with 1 μg/ml anti-CD16/32 antibody and incubated with 2 μg/ml biotin-anti-Ly6G (BioLegend), 2 μg/ml biotin-anti-Ly6C (BioLegend), 2 μg/ml biotin-anti-CD90.2 (BioLegend), or 2 μg/ml biotin-anti-B220 antibodies (BioLegend) at 4 °C for 1 h, followed by anti-biotin microbeads (Miltenyi Biotech) at 4 °C for 1 h. The labeled cells were positively isolated by the magnetic sorting using autoMACS.

**DNA microarray**. DNA microarray analysis was conducted by Chemicals Evaluation and Research Institute (Saitama, Japan). Briefly, RNA integrity and quality were determined using 2100 Bioanalyzer (Agilent Technologies, Wokingham, UK). RNA was reverse-transcribed to cDNA and transcribed into Cy3-labeled cRNA using a Low Input Quick Amp Labeling Kit (Agilent, Santa Clara, CA) according to the manufacturer's instructions. Samples were hybridized to SurePrint G3 Mouse GE Microarray (Ver. 2.0) in an oven. The array was scanned with an Agilent Microarray Scanner (Agilent Technologies). The DNA microarray dataset is shown in Supplementary Data 1.

GO analysis of the genes downregulated in $Tarm1^{-/-}$ GM-DCs more than twofold was performed using PANTHER Overrepresentation Test (release 2017-04-13, data base released 2017-07-21) and adjusted for multiple testing by Bonferroni correction.

**Recall T cell proliferation assay**. Inguinal LNs were isolated from mice sacrificed at day 10 after primary immunization. Single-cell suspensions were prepared from the dLNs and were cultured in the presence of the indicated concentrations of heat-denatured chicken IIC in 96-well plates ($2 \times 10^5$/well). After incubation for 66 h, cells were labeled with [$^3$H]-thymidine (0.25 μCi/ml, PerkinElmer, Waltham, MA) for 6 h and harvested using a Micro 96 cell harvester (Skatron, Sterling, VA). [$^3$H]-radioactivity was measured using a Micro Beta System (Pharmacia Biotech, Piscataway, NJ). Culture supernatants were collected after 66 h simulation.

**Measurement of cytokine production**. Cytokine production were determined using Mouse TNF ELISA MAX (BioLegend), Duo set IFN-γ (R&D systems, Minneapolis, MN), Duo set IL-17 (R&D systems), Cytometric Bead Array (CBA) mouse TNF Flex Set (BD Biosciences), CBA mouse IL-6 Flex Set (BD Biosciences), and CBA mouse IL-10 Flex Set (BD Biosciences) in accordance with the manufacture's protocols.

**Cell stimulation**. GM-DCs or BM neutrophils ($2 \times 10^5$/well) were incubated in the presence of *Escherichia coli* O111:B4 LPS (100 ng/ml; SIGMA-ALDRICH), CpG (10 μM; Operon, Eurofins Genomics, Tokyo, Japan), poly(I:C) (10 μM; InvivoGen, San Diego, CA), TNF (10 ng/ml; Peprotech), IL-1β (10 ng/ml; Peprotech), zymosan (100 μg/ml; SIGMA-ALDRICH), *M. tuberculosis* H37 Ra (100 μg/ml), and IIC (1, 10, and 100 μg/ml; SIGMA-ALDRICH) at 37 °C for the indicated time. T or B cells ($2 \times 10^5$/well) were stimulated with either plate-bound anti-CD3 (1 μg/ml; clone, 145-2C11; eBioscience) or soluble anti-IgM antibodies (10 μg/ml; Jackson ImmunoResearch), respectively.

**OVA uptake**. GM-DCs ($2 \times 10^6$/ml) were suspended with RPMI1640 containing 2% FBS and incubated with 0.1 mg/ml AlexaFluor 647 (AF647)-OVA (Invitrogen) for the indicated time points at 37 °C. After washing twice with ice-cold RPMI1640/2% FBS, cells were stained with fluorescence-conjugated antibodies and analyzed by flow cytometry. Mean fluorescence intensity and frequency of AF647-positive cells were calculated.

**Plasmids**. For construction of pcDNA3.1/HA vector, a fragment encoding HA sequence was ligated into pcDNA3.1 (+) vector (Invitrogen) via *Xho*I and *Xba*I sites. The full-length cDNA fragment of TARM1 (NM_177363.3; 1–288 aa) was obtained from a BM cDNA library by PCR and cloned into pcDNA3.1/HA vector via *Nhe*I and *Eco*RV sites. A gene encoding an extracellular region of TARM1 (22-254 aa) was amplified by PCR with a primer set (Supplementary Table 1) and inserted into pFUSE-hIgG2-Fc2 vector (Invivogen) via *Eco*RV and *Nco*I sites. A gene encoding a Flag-Thrombin sequence (CCATGGGTAGTAGTGACTACAAAGACGATGACGACAAGCTGGTGCCGCGCGGTAGTAGTACCATGG) was inserted into the pFUSE-hIgG2-Fc2/TARM1 ECD via *Nco*I site.

**Preparation of TARM1-Fc, IgG Fc, and TARM1-Flag**. 293T cells were kindly provided by Professor Hiroaki Tateno (National Institute of Advanced Industrial Science and Technology). Plasmids were transfected into 293T cells by Lipofectamine LTX (Invitrogen) in accordance with the manufacturer's instruction. After incubation for 24 h, culture media were changed to 2% low IgG FBS (Invitrogen) in OPTI-MEM (Invitrogen). Fusion proteins secreted into medium were purified by affinity chromatography on Protein A-Sepharose. For preparation of TARM1-Flag, TARM1-Flag-Fc was treated with 10 U Thrombin (Nacalai Tesque), and then TARM1-Flag was purified by chromatography on Protein A-Sepharose followed by Benzamidine-Sepharose.

**Solid-phase-based binding analysis**. Maxisorp 96-well plates (ThermoFisher SCIENTIFIC) were coated with IC, IIC, and BSA (0–10 μg/ml), washed with wash buffer (TBS containing 0.05% Tween20, 2 mM $CaCl_2$) and blocked with blocking buffer (TBS containing 0.05% Tween20 and 5% BSA). Fc-fused proteins (5 μg/ml) in binding buffer (TBS containing 0.05% Tween20, 2 mM $CaCl_2$, and 0.5% BSA) were incubated at room temperature for 2 h. Horseradish peroxidase (HRP)-conjugated goat anti-human IgG (Fcγ fragment specific; 1:5000 dilution; Jackson ImmunoResearch, West Grove, PA; cat# 109-035-098; lot# 128332) was incubated at room temperature for 1 h. SureBlue TMB Microwell Peroxidase Substrate (SeraCare, Milford, MA) was added to each well, and the reaction was stopped with 1 N HCl. Color development was monitored at 450 nm with an iMark microplate reader (BIO-RAD, Hercules, CA).

**Immunoprecipitation with TARM1-Fc and Western blotting**. Cells were lysed with TBS containing 1% NP-40 and a Protease Inhibitor Cocktail for Use with Mammalian cell and Tissue Extracts (Nacalai Tesque). After preclearing, cell lysate was incubated with TARM1-Fc- and IgG Fc (10 μg)-immobilized Protein A (10 μl)-Sepharose at 4 °C overnight. After washing, the slurry was boiled at 100 °C for 10 min with sodium dodecyl sulfate (SDS) sample buffer. The samples were centrifuged, and supernatants were subjected to SDS-polyacrylamide gel electrophoresis analysis followed by electroblotting analysis onto a polyvinylidene difluoride membrane. The membrane was blocked with 4% BlockAce (DS Pharma Biomedical Co., Osaka, Japan) and then incubated with 1 μg/ml biotin-conjugated

anti-IIC antibody (clone, A2-10, F10-21, D8-6 and D1-2G; Chondrex, Redmond, WA; cat# 7006; lot# 150116) followed by HRP-conjugated Streptavidin (1:1000 dilution; BioLegend; cat# 405103; lot# 79004). Whole-cell lysate was also immunoblotted with anti-IIC (1 μg/ml) and anti-β-actin antibodies (1:5000 dilution; MBL, Aichi, Japan; cat# PM053; lot# 005). The membrane was incubated with ECL Prime Western Blotting Detection System (GE Healthcare, Little Chalfont, England), and chemiluminescent signal images were acquired using a digital imager, LAS4000 system (Fuji Film Life Science, Tokyo, Japan).

**Blocking of GM-DC activation with soluble TARM1**. IIC (1 μg/ml) was precoated on 96-well plates. After washing of the plates with PBS, TARM1-Fc, Fc, and TARM1-Flag (1, 3, and 10 μg/ml) were incubated for 1 h at 37 °C. Then GM-DCs at day 8 after GM-CSF treatment ($2 \times 10^5$/well) were added to the plate and incubated for 24 h at 37 °C. The cells were collected using Cell Dissociation Solution (SIGMA-ALDRICH), and cell activation was analyzed by flow cytometry.

**T cell differentiation**. CD4$^+$ T cells were isolated from the spleen and LN using autoMACS with anti-CD4-conjugated microbeads (Miltenyi Biotech). For Th1 polarization, cells were stimulated with plate-coated anti-CD3 antibody (4 μg/ml; clone, 145-2C11; eBioscience), soluble anti-CD28 antibody (1 μg/ml; clone, 37.51; eBioscience), anti-IL-4 antibody (10 μg/ml; clone, 11B11; eBioscience) and IL-12 (10 ng/ml; Peprotech) for 3 days. For Th17 polarization, cells were stimulated with anti-CD3 antibody (4 μg/ml), anti-CD28 antibody (1 μg/m), anti-IFN-γ antibody (10 μg/ml; clone, XMG1.2; eBioscience), anti-IL-4 antibody (10 μg/ml), TGF-β (3 ng/ml; Peprotech), IL-6 (40 ng/ml; Peprotech), IL-1β (20 ng/ml; Peprotech), and IL-23 (20 ng/ml; R&D systems) for 3 days.

**Statistics**. Two-tailed unpaired Student's $t$ test, $\chi^2$-test, and two-tailed Mann–Whitney $U$ test are used for statistical evaluation. $P$ value <0.05 is defined as significant; *$P$ < 0.05; **$P$ < 0.01; ***$P$ < 0.001. Data are expressed as mean ± SD as indicated.

**Reporting summary**. Further information on research design is available in the Nature Research Reporting Summary linked to this article.

## Data availability

All data supporting the results reported here are available in the article and Supplementary Information files or from the corresponding author upon reasonable request. Source data are provided with this paper.

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

## Acknowledgements

This work is supported by the Science and Technology Research Promotion Program for Agriculture, Forestry, Fisheries and Food Industry (to Y.I.), Grand-in-Aid for Scientific Research from the Ministry of Education, Science and Culture of Japan (24220011 to Y.I.), and Grant-in-Aid for JSPS Fellows (11J09956 to R.Y.).

## Author contributions

R.Y. designed, performed experiments, analyzed data, and wrote the manuscript; S.H. and M.A.M. contributed the CIA study; S.K. provided technical support for the generation of *Tarm1*−/− mice; Y.A., T.M., A.S., and T.K. assisted in vitro study and provided crucial advice; K.S. assisted with DNA microarray analysis; S.S. gave crucial advice and technical support; and Y.I. organized, supervised the research project, and edited the manuscript.

## Competing interests

The authors declare no competing interests.
