## [Peer Review File · Nature Communications]

Reviewers' comments:

Reviewer #1 (Remarks to the Author):

Yabe et al report that TARM1 plays an essential role in arthritis development through binding to collagens leading to activation of dendritic cells. The authors explored Tarm1^{-/-} mice to demonstrate its implication in arthritis development. Moreover, T cell priming against type 2 collagen was suppressed probably due to an impairment of antigen-presenting cells. Next, the authors found that TARM1 binds to type 2 collagen and that soluble fusion protein TARM1-Fc inhibited DC maturation and reversed collagen-induced arthritis. These observations are clear-cut and if confirmed in humans can provide new therapeutic options for rheumatoid arthritis. However, there are some concerns that need to be solved before these conclusions can be validated.

Comments:

1. By which means TARM1-Fc is binding to Tarm1^{-/-} CD11c⁺ cells? Are FcRs involved?
2. Binding experiments need to be done without the Fc portion, with tags for example.
3. The authors need to use soluble TARM1 without Fc portion to evaluate its direct effect and compared with other related proteins such as soluble OSCAR. Although the authors have used IgG Fc as control for binding and for treatment, they cannot exclude the role of Fc receptors in this inhibitory effect. FcγRIIb (CD32b) that display ITIM motif can be involved (Nimmerjahn & Ravetch Nat Rev Immunol 2008). In addition, FcγRIII (CD16) that is also associated with FcR ITAM, can mediate inhibition through ITAM motif when dimerized (Aloulou et al Blood 2012). As mentioned by the authors, TARM1-Fc makes complexes with type 2 collagen and such complexes can engage such FcRs delivering inhibitory signals that differs from IgG Fc.

Reviewer #2 (Remarks to the Author):

The main point of this paper, that dendritic cells express (or acquire?) type II collagen that serves as the ligand for Tarm1, and subsequently inhibits the development of autoimmune arthritis, is weakly supported by the data. Significantly more data needs to be provided to convince the reader that dendritic cells express type II collagen, or at least somehow acquire it on its surface, as well as evidence that Tarm1^{-/-} mice are immunologically sufficient.

That said, the authors' inclusion of more than 100 graphs and pictures in Figures 1-6, and more than 50 in their supplementary data simply adds more confusion to the manuscript than aids it. They attempt to describe so many experiments in such a limited narrative space, that there is little detail or thought-provoking narrative provided for each of them. Additionally, much of the excessive data detract from the intended focus of this manuscript. The result is a confusing set of data that is nearly impossible to follow or interpret, and the reader is apparently expected to readily accept.

Additionally, the extensive reliance of the authors on statistical significance over biological significance is problematic. Repeatedly, very small differences in measurements are shown to be statistically significant with no discussion whatsoever on how these very small differences would result in biological differences, and it is highly doubtful that they do. The net effect is that these data fail to convince the reader of the significance of the findings.

Some issues of note:

- Authors repeatedly describe their new therapy as curing disease. It clearly does not. It may attenuate the disease, but disease is still clearly present.
- R-squared analyses on data of only 3 points is questionable (Figure 1c and 1d)

- A careful analysis of the immune status of the Tarm1^{-/-} mice should be provided
- There are issues with the CIA data. Several Tarm1^{-/-} mice have same level of anti-CII antibody as mice in the positive control group, yet these Tarm1^{-/-} mice did not develop disease. Why? If the CII-specific T cell response is inhibited in these mice, how did the antibody titers reach levels similar to those in the mice from the control group that did develop arthritis? This is especially a problem given that antibody is the primary mechanism of pathogenesis in this model.
- The focus on GFP expression in the Tarm1^{-/-}KO mice as an indicator of presumed protein expression of Tarm1 is not appropriate. Insertion of the GFP gene in the Tarm1 gene could easily alter transcription rates, levels, etc.

Reviewer #3 (Remarks to the Author):

In this manuscript, Yabe et al explore the role of TARM1, which associates with FcR-gamma for macrophage and neutrophil activation and which is a gene over-expressed in common in 2 models of inflammatory arthritis. The authors generate a Tarm1 reporter/ko mouse and demonstrate reduced disease severity. TARM1 is highly expressed in monocyte-derived Ly6c⁺ monocyte-derived DC, and the ko mouse has fewer class IIhi DC. In cultures of T cells and DC from mice with CIA, they find T cell proliferation in response to collagen II, but not with TARM1ko DC. TARM1 expression increases with differentiation of BM GM-DCs and ko GM-DCs are less activated than wt. TARM1-Fc blocked this activation. However, osteoclast precursors could still differentiate into osteoclasts with M-CSF and RANKL. Endogenous activation was blocked by collagenase, and IC and IIC bound TARM1 to activate DC. Both ko and wt DC responded to TLR agonists, while ko BM neutrophils made a lower cytokine response to TLR agonists than wt neutrophils. TARM1-Fc administration reduced CIA severity, relative to IgG-Fc, associated with increased IL-10 and reduced MPO transcription.

Novelty and interest: the work is novel and potentially of interest if technically sound. The effect of TARM1-Fc on arthritis severity is impressive, suggesting TARM1 is, as claimed, a potentially interesting target. However, I did not find the conclusions completely convincing with regard to DC mechanism, the claim that IIC is an endogenous/natural ligand of TARM1 or the impact on T cell responses. I have outlined these concerns below including potential technical issues.

Specific points:

1. the amount of IIC antigen needed to elicit a T cell proliferative or cytokine response in vitro is very high (at least 50mg/ml) and the response is measured using thymidine incorporation, suggesting the proliferative/cytokine response may not derive from T cells but could derive e.g. from DCs in the culture (given that IIC is an activating ligand for DC). These assays should be redone using CFSE labelling and i.c. cytokine production of cells to identify the proliferating and cytokine-producing cells. If the IIC class II epitope for C57BL6 is known it could be tested in vitro (as T cells should be much more sensitive to low concentrations of peptide). Furthermore, the proliferative response may not be due to collagen antigen presentation in this assay but non-specific stimulation by the high levels of added IIC. An antigen presentation assay should be carried out to measure response to a control antigen e.g. prime to OVA rather than IIC, assay endogenous OVA peptide-specific T cell proliferation using wt or ko DC in the presence or absence of IIC.
2. Figure 3: what is TARM1 expression on pDC (siglec H⁺)?
3. The reduction in cytokine production by ko BM neutrophils in response to TLR agonists does not support the overall hypothesis that lack of stimulation of GM-DC TARM1 by the putative CII ligand is the mechanistic explanation for the reduced inflammation in the ko mice. Indeed reduction in MPO after administration of TARM1-Fc suggests a major impact on neutrophils.
4. Figure 4e: anti-IC and IIC antibodies are used to identify binding of IC and IIC to the surface of GM-DCs. No information is given about these antibodies – were they polyclonal and what is their specificity especially in flow assays? Where could the collagen ligand have come from in these experiments? My understanding is that IIC is only expressed in joint cartilage and the sclera of the eye. Does TARM1-Fc bind to other members of the collagen family?

Reviewers' comments:

Reviewer #1 (Remarks to the Author):

Thank you very much for your time and the precious comments. We describe the responses to the comments below.

Yabe et al report that TARM1 plays an essential role in arthritis development through binding to collagens leading to activation of dendritic cells. The authors explored Tarm1^{-/-} mice to demonstrate its implication in arthritis development. Moreover, T cell priming against type 2 collagen was suppressed probably due to an impairment of antigen-presenting cells. Next, the authors found that TARM1 binds to type 2 collagen and that soluble fusion protein TARM1-Fc inhibited DC maturation and reversed collagen-induced arthritis. These observations are clear-cut and if confirmed in humans can provide new therapeutic options for rheumatoid arthritis. However, there some concerns that need to be solved before these conclusions can be validated.

1. By which means TARM1-Fc is binding to Tarm1^{-/-} CD11c⁺ cells? Is FcRs involved?

Response: Because we used anti-CD16/CD32 antibody (clone 2.4G2), which is commonly used to prevent non-specific binding to FcRs, to prevent non-specific binding of Fc-fused proteins to FcRs, we think that the binding of Fc-fused proteins to FcRs is negligible.

2. Binding experiments need to be done without the Fc portion, with tags for example.

Response: We used IgG Fc as a background control, but we did not detect any binding (Fig. 4a). These data clearly indicate that FcRs were satisfactory blocked by anti-CD16/CD32 antibody, and TARM1-Fc binding to CD11c⁺ cells was not mediated by FcRs. Thus, we think additional experiments are not necessarily required.

3. The authors need to use soluble TARM1 without Fc portion to evaluate its direct effect and compared with other related proteins such as soluble OSCAR. Although the authors have used IgG Fc as control for binding and for treatment, they cannot exclude the role of Fc receptors in this inhibitory effect. FcγRIIb (CD32b) that display ITIM motif can be involved (Nimmerjahn & Ravetch Nat Rev Immunol 2008). In addition, FcγRIII (CD16) that is also associated with FcR ITAM, can mediated inhibition through ITAM motif when dimerized (Aloulou et al Blood 2012). As mentioned by the authors, TARM1-Fc makes complexes with type 2 collagen and such complexes can engage such FcRs delivering inhibitory signals that differs from IgG Fc.

Response: Thank you for the important comment. According to your comments, we examined the effects of TARM1-Flag instead of TARM1-Fc on the suppression of type 2 collagen-induced activation of DCs. We showed that TARM-1-Flag also can inhibit the induced activation, indicating that the suppression was caused by the TARM1 portion, but not

by the Fc portion, of the molecule. These results are added to the figures as new Fig. 4i and we described in the text (page 11, line 10). It was difficult to produce TARM1 without Fc or Flag because we have no appropriate purification methods.

As the reviewer pointed out, stimulation of Fc γ RIIb and Fc γ RIII by IgG Fc is known to introduce negative signaling. However, this requires high concentrations of IgG (> 5 mg/ml) (Aloulou M et al., *Blood*, 119, 3084-3096, 2012). In our experiments shown in the original Figs. 5k and l, the concentration of TARM1-Fc was 10 μ g/ml, and only 1 μ g was injected to a mouse to treat CIA in the experiments shown in new Fig. 5. Thus, the Ig concentrations in our experiments were not enough to activate these inhibitory Fc receptors. Also, previous reports indicated that human IgG Fc have no therapeutic effects on arthritic joints (Hah YS et al., *Arthritis Res Ther*, 15, R85, 2013; Zhu W et al., *Inflammation*, 39, 839-848, 2016; Zhou B et al., *Signal Transduct Target Ther*, 4, 19, 2019). Ricks et al. also showed that expression of mIgG1 Fc or mIgG2a Fc did not significantly suppress fungus-induced inflammatory responses, although Dectin-1:mIgG1 and Dectin-1:mIgG2a can suppress the responses through Dectin-1 (Ricks DM et al., *Infect Immun*, 81, 3451-3462, 2013; Fig. 6C). Thus, we think that the observed effects of TARM1-Fc are caused by TARM1, and Fc portion is not involved in the response under our experimental conditions.

Reviewer #2 (Remarks to the Author):

Thank you very much for your time and the important comments. We describe the responses to the comments below.

The main point of this paper, that dendritic cells express (or acquire?) type II collagen that serves as the ligand for Tarm1, and subsequently inhibits the development of autoimmune arthritis, is weakly supported by the data. Significantly more data needs to be provided to convince the reader that dendritic cells express type II collagen, or at least somehow acquire it on its surface, as well as evidence that *Tarm1*^{-/-} mice are immunologically sufficient.

Response: Thank you for the important comments. However, we showed that *Col2a1*, but not *Col1a1*, mRNA is expressed in GM-DCs, as compared with mouse embryonic fibroblasts (MEFs) (new Fig. 4c). Thus, it is clear that type 2 collagen detected on GM-DC surface was not acquired from other cells but directly produced in these cells. We add this in the Results section (page 10, line 18).

Regarding immunological sufficiency of *Tarm1*^{-/-} mice, these mice looked healthy under SPF conditions and their lymphatic organs looked normal. We have newly examined the immune cell compositions, such as DCs, Mfs, Mos, Neus, B, CD4⁺ T and CD8⁺ T cells, in the spleen, LNs and BM, and found that they are quite normal. These results are described in the text (page 6, line 13) and new Supplementary Fig. 1h. Furthermore, we already showed that *Tarm1*^{-/-} T cell- and B cell-proliferation after anti-CD3 and anti-IgM treatment, respectively, was normal (Supplementary Fig. 2g, h), and Th1- and Th17-differentiation ability were normal in *Tarm1*^{-/-} CD4⁺ T cells (Supplementary Fig. 2i, j). Furthermore, phagocytic activity (Supplementary Fig. 2f) and cytokine production after treatment of LPS, CpG, Poly(I:C), and Zymosan were normal in *Tarm1*^{-/-} GM-DCs (Supplementary Fig. 3g). However, gene expression in GM-DCs after induction of differentiation of BM cells with GM-CSF (Fig. 2g), recall T cell memory responses of T cells from CIA-induced mice (Fig. 3a-d), antigen-presenting ability of GM-DCs to OT-II mouse T cells (Fig. 3h) and T cell proliferation in mixed lymphocyte culture (Fig. 3i), were significantly diminished in cells from *Tarm1*^{-/-} mice. Because these defective functions found in *Tarm1*^{-/-} cells depend on mature DCs, we concluded that DC maturation is suppressed in *Tarm1*^{-/-} mice.

That said, the authors inclusion of more than 100 graphs and pictures in Figures 1-6, and more than 50 in their supplementary data simply adds more confusion to the manuscript than aids it. They attempt to describe so many experiments in such a limited narrative space, that there is little detail or thought-provoking narrative provided for each of them. Additionally, much of the excessive data detract from the intended focus of this manuscript. The result is a confusing set of data that is nearly impossible to follow or interpret, and the reader is apparently expected to readily accept.

Response: Thank you for your important suggestion. Accordingly, we have moved many

unessential Figures into Supplementary Figs (original Figs. 1a, b, e-g; Fig. 3f; Figs. 4b, d, e, i-k, m; and Fig. 5j) and deleted Figs and Supplementary Figs (original Figs. 1c, d; Figs. 5a, d; original Supplementary Fig. 1c; Supplementary Figs. 2a-c, e, f, k-t; Supplementary Figs. 3a-d; and Supplementary Figs. 4f, g) to focus the defects of DC function in *Tarm1*^{-/-} mice. I think the manuscript is now much readable and easy to understand.

Additionally, the extensive reliance of the authors on statistical significance over biological significance is problematic. Repeatedly, very small differences in measurements are shown to be statistically significant with no discussion whatsoever on how these very small differences would result in biological differences, and it is highly doubtful that they do. The net effect is that these data fail to convince the reader of the significance of the findings.

Response: We are confident about our data. For example, the differences of mature DC cell population between WT and *Tarm1*^{-/-} mice, as examined by CD86 and I-A/I-E expression, are relatively small (Fig. 3e, f). However, they are statistically significant and reproducible. We think this is because I-A or CD86 represents only a part of molecules which are involved in T cell activation. As shown in Fig. 2g, many gene expressions were decreased in *Tarm1*^{-/-} mice, and probably these genes are coordinately involved in antigen presentation and T cell activation in a synergistic or additive manner. Thus, sum of these small changes can explain the big difference of T cell response shown in Fig. 3a-d, g-i and Fig. 4g, h, which clearly indicate the defect of DCs to help T cell activation and to produce cytokines upon stimulation with type II collagen. We add this in the discussion section (page 14, line 6).

Some issues of note:

- Authors repeatedly describe their new therapy as curing disease. It clearly does not. It may attenuate the disease, but disease is still clearly present.

Response: Thank you for the comment. We have corrected the text throughout the manuscript.

- R-squared analyses on data of only 3 points is questionable (Figure 1c and 1d)

Response: Because reduction of experimental animals is an important principle for animal experiments, we used the minimum number of animals. However, we have removed original Figs. 1c, d in the revised manuscript because these data are not essential in this manuscript.

- A careful analysis of the immune status of the *Tarm1*^{-/-} mice should be provided

Response: As described above, we have newly analyzed the immune cell compositions in the bone marrow, lymph nodes and spleen from WT and *Tarm1*^{-/-} mice under the physiological conditions (new Supplementary Fig. 1h), in addition to intrinsic T cell and B cell function (Supplementary Figs. 2g, h), Th1- and Th17-differentiation (Supplementary Figs. 2i, j), recall T cell memory responses of CIA-induced mice (Fig. 3a-d), antigen-presenting ability of

GM-DCs to OT-II mouse T cells (Fig. 3h), and mixed lymphocyte culture (Fig. 3i), gene expression in GM-DCs (Fig. 2g), and cytokine production in activated DCs (Supplementary Fig. 3g). No significant difference of immune cell compositions between WT and *Tarm1*^{-/-} littermates was detected (new Supplementary Fig. 1h). These results were described the text (page 6, line 13).

- There are issues with the CIA data. Several *Tarm1*^{-/-} mice have same level of anti-CII antibody as mice in the positive control group, yet these *Tarm1*^{-/-} mice did not develop disease. Why? If the CII-specific T cell response is inhibited in these mice, how did the antibody titers reach levels similar to those in the mice from the control group that did develop arthritis? This is especially a problem given that antibody is the primary mechanism of pathogenesis in this model.

Response: Thank you for the critical comment. We showed that the average levels of IgG2a, IgG2b and IgG3, but not of IgG1, anti-type II collagen were significantly decreased in *Tarm1*^{-/-} mice compared with WT mice (new Fig. 1h). This is consistent with our notion that T cell priming activity of DCs is reduced in *Tarm1*^{-/-} mice. However, the antibody levels were not correlated exactly with the arthritic severity as pointed out by the reviewer; some non-arthritic mice in *Tarm1*^{-/-} group also produced high levels of antibodies. We previously also reported that serum IgG rheumatoid factor (RF) levels does not absolutely correlate with the development of arthritis in a rheumatoid arthritis (RA) model (HTLV-I transgenic mice) (Iwakura Y et al., Science, 253, 1026-1028, 1991; Iwakura Y et al., J Immunol, 155, 1588-1598, 1995), although collagen-induced arthritis susceptible mice are high antibody responders (Wooley PH et al., J Exp Med, 154, 688-700, 1981). Consistent with these observations, Campbell et al. also reported that anti-IIC antibody levels do not correlated with incidence of CIA in mice (Campbell IK et al., Eur J Immunol., 30, 1568-1575, 2000). Elevated levels of IgG RF are observed in patients with RA, but several reports indicated that serum RF levels are elevated in healthy individuals and non-RA patients (Linker JB III et al., Man Clin Lab Immunol, 1986; Ingegnoli F et al., Dis Markers, 35, 727-734, 2013). The reason why antibody levels in serum are not absolutely corresponded to the arthritic severity is not known completely, but these observations may suggest that all the antibodies against IgG are not necessarily pathogenic in the development of arthritis. Also, other autoantibodies such as anti-citrullinated peptides may also be involved in the pathogenesis. Actually, there are multiple non-arthritisogenic epitopes which can induce high levels of antibodies other than arthritisogenic epitopes on the type II collagen molecules (Wooley PH et al., J Immunol, 135, 2443-3451, 1985). It is also possible that pathogenic antibodies are trapped by the Ag in joints (Vaughan JH, Arthritis Rheum, 36, 1-6, 1993). Thus, definite answer to this question is not possible at present, but obviously, this is beyond the scope of this research.

We have renamed Fig. 1j into Fig. 1h, and described the reason in the Introduction and Results sections (page 5, line 9; page 7, line 23).

- The focus on GFP expression in the *Tarm1*^{-/-}KO mice as an indicator of presumed protein expression of *Tarm1* is not appropriate. Insertion of the GFP gene in the *Tarm1* gene could easily alter transcription rates, levels, etc.

Response: Thank you for the important comment. We have analyzed *Tarm1* expression in different cells using qPCR (Fig. 2c), and we confirmed that *Tarm1* is highly expressed in GM-DCs and at lower levels in BM-Mfs and BM-Neus (new Supplementary Fig. 2b), in consistent with the results obtained by FACS analysis using GFP as the marker. Therefore, we think in this case we can use GFP expression to trace *Tarm1* expression. Consistent with our results, Radjabova et al. demonstrated that *Tarm1* is expressed in BM-Mfs and BM-Neus, but not in T and B cells (Radjabova et al., J Immunol, 2015). Therefore, we think GFP expression faithfully represents *Tarm1* expression. We added this information in the Result (page 7, line 15) and the data are added as new Supplementary Fig 2a and b.

Reviewer #3 (Remarks to the Author):

Thank you very much for your time and the precious comments. We describe the responses to the comments below.

1. the amount of IIC antigen needed to elicit a T cell proliferative or cytokine response in vitro is very high (at least 50mg/ml) and the response is measured using thymidine incorporation, suggesting the proliferative/cytokine response may not derive from T cells but could derive e.g. from DCs in the culture (given that IIC is an activating ligand for DC). These assays should be redone using CFSE labelling and i.c. cytokine production of cells to identify the proliferating and cytokine-producing cells. If the IIC class II epitope for C57BL6 is known it could be tested in vitro (as T cells should be much more sensitive to low concentrations of peptide). Furthermore, the proliferative response may not be due to collagen antigen presentation in this assay but non-specific stimulation by the high levels of added IIC. An antigen presentation assay should be carried out to measure response to a control antigen e.g. prime to OVA rather than IIC, assay endogenous OVA peptide-specific T cell proliferation using wt or ko DC in the presence or absence of IIC.

Response: We sincerely apologize our mistakes in the units of the original Fig. 2a-d x-axis (now Fig. 3a-d). We used not up to 200 mg/ml but up to 200 microgram/ml IIC in the proliferation assay. Therefore, DCs do not efficiently proliferate under our experimental conditions. Furthermore, the DC number in the draining lymph nodes was very small compared with T and B cells. Thus, we think that T cells, but not DCs, mainly contributed to the cytokine production and [³H]-thymidine incorporation in the proliferation assay under our experimental conditions.

We also showed that antigen-presenting ability of *Tarm1*^{-/-} GM-DCs is diminished in a co-culture system with CD4⁺ T cells from OT-II Tg mice (Fig. 3h) and in a mixed lymphocyte culture with BALB/cA T cells (Fig. 3i). These results clearly indicate that antigen-presenting ability of *Tarm1*^{-/-} GM-DCs is diminished not only against IIC but also against other antigens.

2. Figure 3: what is TARM1 expression on pDC (siglec H+)?

Response: We defined B220⁺CD11c⁺ cells as pDCs according to Swiecki M and Colonna M. (Nat Rev Immunol, 15, 471-485, 2015), and we detected only very low expression of *Tarm1* in these cells (new Supplementary Fig. 2a).

3. The reduction in cytokine production by ko BM neutrophils in response to TLR agonists does not support the overall hypothesis that lack of stimulation of GM-DC TARM1 by the putative CII ligand is the mechanistic explanation for the reduced inflammation in the ko mice. Indeed reduction in MPO after administration of TARM1-Fc suggests a major impact on neutrophils.

Response: Thank you for your important comment. As you pointed out, since cytokine

production was reduced in *Tarm1*^{-/-} BM-derived neutrophils following stimulation with TLR and CLR agonists, neutrophils in *Tarm1*^{-/-} mice may have some defects and it is possible that defects of neutrophils affect the development of CIA. However, the defects in neutrophils cannot explain completely the defects observed in *Tarm1*^{-/-} mice because neutrophils are not involved in the T cell sensitization and antibody production. Given that Tarm1 is most highly expressed in GM-DCs among bone marrow-derived cells (Fig. 2c), and TARM1 controls antigen presentation and cytokine production of DCs upon stimulation with IIC, we think that defects in DCs mainly contribute to the defect of CIA development in *Tarm1*^{-/-} mice, and the reduction in MPO after administration of TARM1-Fc is a result of inflammation suppression. We have added these sentences in the Discussion section (page 15, line 16).

4. Figure 4e: anti-IC and IIC antibodies are used to identify binding of IC and IIC to the surface of GM-DCs. No information is given about these antibodies – were they polyclonal and what is their specificity especially in flow assays? Where could the collagen ligand have come from in these experiments? My understanding is that IIC is only expressed in joint cartilage and the sclera of the eye. Does TARM1-Fc bind to other members of the collagen family?

Response: Thank you for your important comment. We used anti-IC monoclonal antibody (clone 8D4A1) from Chondrex and anti-IIC polyclonal antibody from R&D Systems for flow cytometric analysis. We also validated IIC expression on GM-DCs using anti-IIC monoclonal antibody from ThermoFisher (new Fig. 4d). The information was added in Supplementary Table 2. We confirmed that these antibodies are functional for flow cytometric analysis using collagen-producing osteoblasts as positive controls.

We analyzed *Col1a1* and *Col2a1* mRNA expression in GM-DCs by qPCR, and found that *Col2a1* mRNA, but not *Col1a1* mRNA, was expressed (new Fig. 4c). This is consistent with the results obtained by flow cytometry with anti-IC and IIC antibodies (new Fig. 4d). We added these data as new Fig. 4c, d and in the Result section (page 10, line 18).

REVIEWERS' COMMENTS

Reviewer #1 (Remarks to the Author):

The authors have satisfactorily answer to my requests. This is a great contribution to the field and I have no further comments.

Reviewer #3 (Remarks to the Author):

The authors have addressed my concerns adequately. The conclusions are original and provide interesting insight into the relationship between a new member of the LILR family and CII in arthritis.